# Gene regulatory network structure informs the distribution of perturbation effects

**Matthew Aguirre** [ID]1*, **Jeffrey P. Spence** [ID]2*, **Guy Sella** [ID]3,4*,
**Jonathan K. Pritchard** [ID]2,5*

**1** Department of Biomedical Data Science, Stanford University, Stanford, California, United States of America, **2** Department of Genetics, Stanford University, Stanford, California, United States of America, **3** Department of Biological Sciences, Columbia University, New York, New York, United States of America, **4** Program for Mathematical Genomics, Columbia University, New York, New York, United States of America, **5** Department of Biology, Stanford University, Stanford, California, United States of America

* magu@stanford.edu (MA); jspence@stanford.edu (JPS); gs2747@columbia.edu (GS); pritch@stanford.edu (JKP)

**Data availability statement:** All relevant data are within the manuscript and its Supporting information files.

## Abstract

Gene regulatory networks (GRNs) govern many core developmental and biological processes underlying human complex traits. Even with broad-scale efforts to characterize the effects of molecular perturbations and interpret gene coexpression, it remains challenging to infer the architecture of gene regulation in a precise and efficient manner. Key properties of GRNs, like hierarchical structure, modular organization, and sparsity, provide both challenges and opportunities for this objective. Here, we seek to better understand properties of GRNs using a new approach to simulate their structure and model their function. We produce realistic network structures with a novel generating algorithm based on insights from small-world network theory, and we model gene expression regulation using stochastic differential equations formulated to accommodate modeling molecular perturbations. With these tools, we systematically describe the effects of gene knockouts within and across GRNs, finding a subset of networks that recapitulate features of a recent genome-scale perturbation study. With deeper analysis of these exemplar networks, we consider future avenues to map the architecture of gene expression regulation using data from cells in perturbed and unperturbed states, finding that while perturbation data are critical to discover specific regulatory interactions, data from unperturbed cells may be sufficient to reveal regulatory programs.

## Author summary

Gene regulatory networks (GRNs) describe the causal relationships by which gene expression is controlled in the cell. How these networks are structured and how this

**Funding:** M.A. acknowledges support from a Microsoft Research PhD Fellowship, and from the National Library of Medicine (https://www.nlm.nih.gov) under training grant T15LM007033. This work was supported by the National Human Genome Research Institute (https://www.genome.gov) under grants R01HG008140 and U01HG012069 (J.K.P.) and by the National Institute of General Medical Sciences (https://www.nigms.nih.gov) under grant R01GM115889 (G.S.). The funders had no role in study design, data collection and analysis, decision to publish, or preparation of the manuscript.

**Competing interests:** The authors have declared that no competing interests exist.

organization relates to their function is a central problem in biology. Here, we propose a framework for simulating GRNs that consists of two parts: an algorithm to create graph structures and a mathematical model of gene expression. We characterize the effects of the parameters of our model, showing how they affect properties of networks and experimental data. Specifically, we find that key structural properties of biological networks–sparsity, modular groups, and degree dispersion–are consistent with patterns in real data and tend to dampen the effects of gene perturbations. We showcase the utility of our model with vignettes from a synthetic network that has properties similar to those of an experimentally assayed GRN. We then describe challenges and opportunities for inference, using graph properties as a lens to consider ways to map the causal and functional relationships between genes in future work.

## 1. Introduction

In the past decade, single cell sequencing assays have been instrumental in enabling functional studies of gene regulatory networks (GRNs). Observational studies of single cells have revealed substantial diversity and heterogeneity in the cell types that comprise healthy and diseased tissues [1], and molecular models of transcriptional systems have been used to understand the developmental processes involved in maintaining cell state and cell cycle [2,3]. Meanwhile, recent advances in the design of interventional studies, including CRISPR-based molecular perturbation approaches like Perturb-seq [4,5], have been useful for learning the local structure of GRNs around a focal gene or pathway [6,7], discovering trait-relevant gene sets at scale [8], and determining novel functions for poorly characterized genes in a particular cell type [9]. The preponderance of single-cell data in multiple cell types, tissues, and contexts has also fueled a resurgence of interest in the wholesale inference of GRNs, capitalizing on new techniques from graph theory and causal inference [10,11].

In functional genomics, network inference and candidate gene prioritization are typical aims of experimental data analysis. In this setting, it is common to make assumptions about the structure and function of GRNs to enable convenient computation. In particular, linear models of gene expression on directed acyclic graphs (DAGs) have been foundational for studies of GRNs, and this approach to structure learning is well described in the literature [12,13]. Many extensions based on this framework have been proposed, including additional sparsity constraints in the form of regression penalties or low-rank assumptions [14,15]. Analogous techniques have also been used in the algorithmic design of perturbation experiments [16].

Even though convenience assumptions like linearity and acyclicity are rarely seen as limiting in practice, it is important to note that they are not always biologically realistic. Gene regulation is known to contain extensive feedback mechanisms [6], and some regulatory structures (in particular, triangles, like the feed-forward motif [17,18]) are not captured well by low-rank representations of GRNs [19]. Furthermore, biological networks are thought to be well described by directed graphs with hierarchical organization and with a degree distribution that follows an approximate power-law [20–22]. In network inference, it is less common to make explicit use of these properties, though there are notable exceptions [7,23].

Meanwhile, empirical competition-based benchmarking studies of GRN inference have revealed several challenges and in the application of network inference approaches to real experimental data. In these competitions, simple heuristic approaches often perform best, as do techniques that leverage perturbation data, assume network sparsity, or ensemble multiple predictions [10,24,25]. Complexity in regulatory network topology and in gene regulation as

a biological process remains a key obstacle to refining inference approaches and establishing reliable benchmarks [26,27]. Pernicious gaps persist between model performance when using synthetic networks for evaluation and performance when using silver-standard ground truth experimental data [11,24,25].

With these practical considerations in mind, it is worth critically examining assumptions which are (or could be) made about the structure of GRNs. In network theory, there are well-established models of networks with group structure [28,29] and with scale-free topologies [30–32]. The defining feature of directed scale-free graphs is a power-law distribution of node in- and out-degrees: this yields emergent properties including group-like structure and enrichment for structural motifs [18]. Further, most nodes in these graphs are connected to one another by short paths, which is referred to as the "small-world" property of networks [33,34].

Here, we characterize in detail a set of structural properties that we consider to be highly relevant for the study of GRNs. We propose a new algorithm to generate synthetic networks with these properties and formulate a gene expression model to simulate data from them. We use this simulation framework to conduct an array of *in silico* functional genomic studies and characterize the parameter space of our model in light of a recent genome-wide Perturb-seq study [9]. Our results provide intuition about the impact of various graph properties on the susceptibility of networks to perturbations, and on the utility of various experimental data types for inference. We conclude by discussing implications for future efforts to map the architecture of gene regulation and complex traits, with particular emphasis on identifying pairwise regulatory relationships between genes and clustering genes into programs. Our analysis tools are available on github (https://github.com/maguirre1/grn-paper) as a resource for the scientific community.

## 2. Results

### 2.1. Modeling approach

Inspired by previous work from network theory and systems biology, we list what we consider to be key properties of GRNs. We motivate these criteria in light of a recent genome-scale study of genetic perturbations, conducted in an erythroid progenitor cell line (K562) (Fig 1) [9]. To date, this is one of the largest available single-cell and single-gene perturbation datasets in any cell type: the data contain measurements on the expression of 5,530 gene transcripts in 1,989,578 cells, which were subject to 11,258 CRISPR-based perturbations of 9,866 unique genes. Here, we subset these data to 5,247 perturbations that target genes whose expression is also measured in the data (**Methods**). Key network properties are as follow:

1. **GRNs are sparse**: While gene expression is controlled by many variables, the typical gene is directly affected by a small number of regulators. We further expect the number of regulators of any single gene to be much smaller than the total number of regulators in the network. Furthermore, not all genes participate in the regulation of expression: only 41% of perturbations that target a primary transcript have significant effects on the expression of any other gene (Fig 1A).

2. **GRNs have directed edges and feedback loops**: Regulatory relationships between genes are directed, with one gene acting as a regulator and the other as a target gene: this means that "A regulates B" is distinct from "B regulates A". Meanwhile, feedback loops are also thought to be pervasive in gene regulatory networks. A simple case of a feedback loop is bidirectional regulation, which is observed in data: 3.1% of ordered gene pairs have at least a one-directional perturbation effect (i.e., "A affects

A: Perturbation effects are sparse

KO alters expression of any gene (41%)

KO has no effects (59%)

B: Edges are directed and can form loops

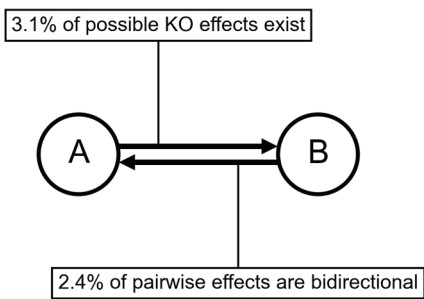

3.1% of possible KO effects exist

A     B

2.4% of pairwise effects are bidirectional

C: Degree distributions are asymmetric

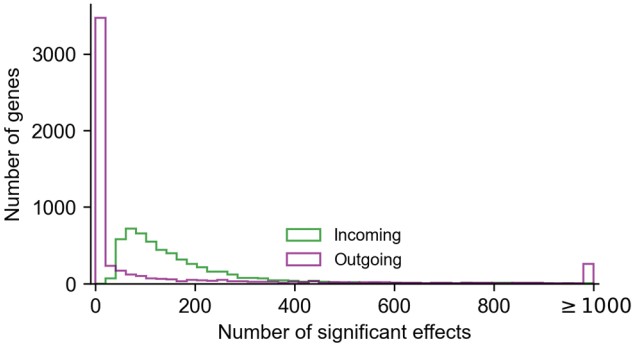

D: Genes are organized into modules

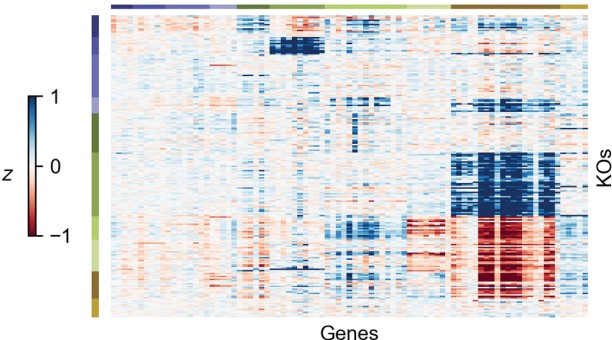

**Fig 1. Key properties of gene regulatory networks**. Data from Replogle et al., 2022. **(A)** Of the 5,247 perturbations in our analysis subset, 2,152 (41%) have a measurable effect on the transcriptional state of cells (energy-test $p < 0.001$). **(B)** Among all ordered pairs of genes, 3.1% (865,719 pairs) have a one-directional effect (FDR-corrected $p < 0.05$). Of these pairs with KO effects, 2.4% (20,621 pairs) further have bidirectional effects. **(C)** Summaries of the distribution of KO effects (Anderson-Darling $p < 0.05$) from the perspective of genes as subject to perturbation (outgoing effects) and as target genes when other genes are perturbed (incoming effects). **(D)** Subset of $z$-normalized expression data corresponding to 10 gene modules, using labels as provided in the dataset—each modules is labeled by a color in the to bars above the $x$- and $y$-axes, and $z$-scores are clipped at $\pm 1$, for visualization.

B", Anderson-Darling FDR-corrected $p < 0.05$), and 2.4% of these pairs further have bi-directional effects (i.e., "B also affects A") (Fig 1B).

3. **GRNs have asymmetric distributions of in- and out-degree**: A further asymmetry between regulators and target genes arises from the existence of master regulators, which directly participate in the regulation of many other genes. The number of regulators per gene and genes per regulator are both thought to follow an approximate power-law distribution [20,21], and indeed, the number of perturbation effects per regulator has a heavier-tailed distribution than the number of effects per target gene (Fig 1C).

4. **GRNs are modular**: Genes in regulatory networks have different molecular functions that are executed in concert across various cell and tissue types. This grouping of genes by function also corresponds to a hierarchical organization of regulatory relationships that is revealed when these programs respond similarly to certain sets of perturbations (Fig 1D).

While these criteria are not exhaustive, they do substantially constrain the space of plausible GRN structures. But from first principles, it is not obvious why GRNs may have these

A: Overview of network generating algorithm

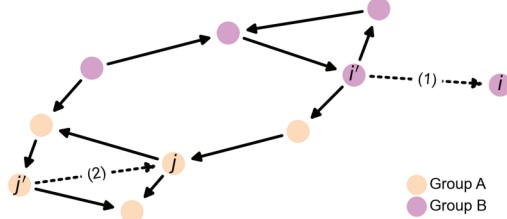

Parameters for 1,920 GRNs in this study:
- Number of genes ($n$) : 2000
- Number of groups ($k$) : $1, \ldots, 100$
- Sparsity term ($p$) : $1/2, \ldots, 1/16$
- In-group term ($w$) : $1, \ldots, 900$
- In-degree term ($\delta_{in}$) : $10, \ldots, 300$
- Out-degree term ($\delta_{out}$) : $1, \ldots, 30$

Perform one of these two steps at random, until the graph has $n$ nodes:

(1). With prob. $p$: Attach a new node ($i$) with an incoming edge. Pick the source ($i'$) using out-degrees ($\delta_{out}$) and groups ($w$).

(2). With prob. $1 - p$: Add a new edge. Pick the target ($j$) using in-degrees ($\delta_{in}$), and the source ($j'$) using out-degrees ($\delta_{out}$) and groups ($w$).

B: Graph properties controlled by algorithm parameters

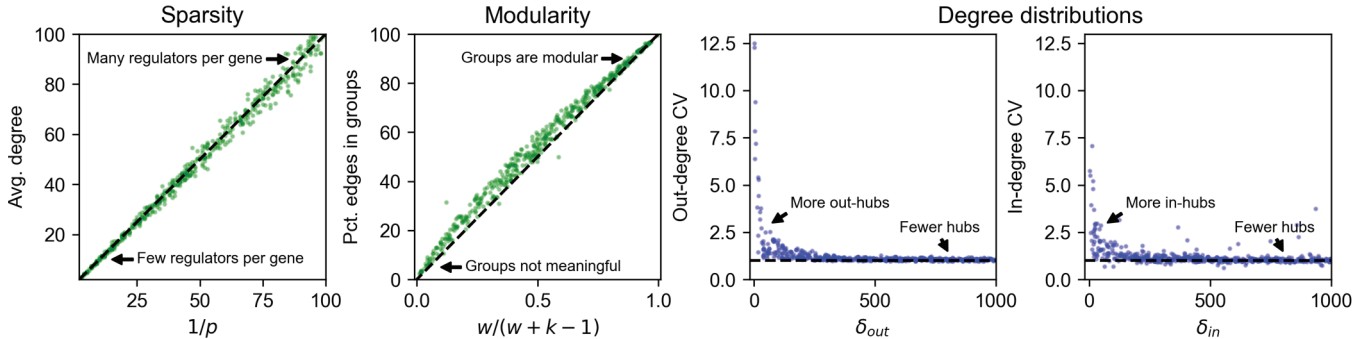

**Fig 2. Modeling approach and network generating algorithm.** **(A)** Overview of network generating algorithm, based on a growth process with preferential attachment. At each step, randomly add either a node or an edge, with the source and target determined by the out- and in-degree distributions, and node membership in groups. **(B)** Key graph properties can be tuned by changing the parameters of the generating algorithm. We validate this in 1,000 synthetic graphs with 500 nodes each, produced with various generating parameters. The same networks are plotted in all four panels, indicating robustness across different background distributions of parameters. Note that in the left two panels, CV is the coefficient of variation of the in- or out-degree distribution, i.e. the mean degree over its standard deviation.

properties, or whether these properties might also limit what can be discovered from experimental data. In other words, what does it matter that networks are sparse, or modular?

## 2.2. Network generating algorithm

To better understand these foundational questions about the impacts of gene regulatory network architecture, we propose a two-step process to simulate synthetic GRNs. First, we produce realistic graph structures using a novel generating algorithm: we show that its parameters control key properties of the resulting graphs. Second, we describe a dynamical systems model of gene expression, which we use to generate synthetic data from arbitrary graph structures. With these tools, we conduct an array of simulated molecular perturbation studies, varying network properties of interest: an overview of our network generating algorithm is in Fig 2A.

Our algorithm is based on that of Bollobás et al., 2003 [32], which models network growth with preferential attachment. This algorithm starts with a small initial graph, and randomly adds nodes or directed edges until the graph reaches a pre-specified size. When adding a node, the new node is selected to be the target of a new directed edge. When adding an edge

between existing nodes, a node is selected to be the target with a probability that increases with the number of outgoing edges it already has. When selecting a node to be the source of a new edge (i.e., to be the regulator for a new gene, if we are adding a node, or for an existing gene, if we are adding an edge), we select with probability increasing in the number of incoming edges it already has. Our work generalizes this algorithm in two ways: first, by assigning each node in the network to one of a number of pre-specified groups, and second, by specifying a within-group affinity term which biases edges to be drawn between members of the same group. The full procedure, including pseudocode and a description of its parameters, is given by Algorithm 1.

**Algorithm 1 Directed scale-free network with groups.**

**Require:**

- $n$: Number of genes (nodes) in the network $(n \geq 3)$.
- $k$: Number of groups in the network $(1 \leq k \leq n)$.
- $p$: Sparsity term $(0 < p \leq 1)$.
- $\delta_{in}$: In-degree biasing term $(\delta_{in} \geq 0)$.
- $\delta_{out}$: Out-degree biasing term $(\delta_{out} \geq 0)$.
- $w$: Group biasing term $(w \geq 1)$.

▷ *Initialize the graph G to be a three-node cycle. Assign each node to its own group.* ◁

$G \leftarrow \{(1 \rightarrow 2), (2 \rightarrow 3), (3 \rightarrow 1)\}$

$\text{gp}(i) \leftarrow i, \quad i \in \{1, 2, 3\}$　　　　▷ *If $k = 2$ then assign node 3 to group 1.*

▷ *Grow the graph G according to the below steps, until it has n genes.* ◁

**while** $|G| < n$ **do**

　▷ *Pick a gene (node) i to be the target of a new regulatory relationship (edge). With probability p, add a new gene to G, otherwise pick an existing gene proportional to the in-degree distribution.* ◁

　**if** $\text{runif}(0,1) < p$ **then**

　　$i \leftarrow |G| + 1$

　　$\text{gp}(i) \leftarrow g \in \{1, ..., k\}$ uniformly at random.

　**else**

　　$i \leftarrow i \in \{1, ..., |G|\}$ with probability $p_i \propto \text{in-degree}(i) + \delta_{in}$

　▷ *Then pick a gene (node) j to regulate i, proportional to the out-degree distribution, weighted by whether i is in the same group as j.* ◁

　$j \leftarrow j \in \{1, ..., |G|\}$ with probability $p_j \propto (\text{out-degree}(j) + \delta_{out}) \times (w$ if $\text{gp}(i) = \text{gp}(j)$ else $1)$

　▷ *Add the edge $(j \rightarrow i)$ to the graph. Note that j and i may be the same node, in which case we add the edge $(j \rightarrow j)$; j and i may also already share the edge $(j \rightarrow i)$, in which case we add a duplicate edge.* ◁

　$G \leftarrow (j \rightarrow i)$

The output of our algorithm is a directed scale-free network on $n$ nodes, each of which is assigned to one of $k$ groups. The parameters in our algorithm control specific network properties. To show this, we generated 1,000 synthetic graphs with $n = 500$ genes using an array of randomly sampled parameters (Fig 2B). We observe that the sparsity term, $p$, adjusts the mean number of regulators per gene, which is approximately $1/p$ (Fig 2B). The number of groups, $k$, and the modularity term, $w$, determine the fraction of edges drawn between members of the $k$ groups – this fraction is approximately $w/(w + k - 1)$ (Fig 2B). Finally, the bias

terms $\delta_{in}$ and $\delta_{out}$ respectively control the coefficient of variation (CV) of the in- and out-degree distributions (Fig 2B and S1 Fig). CV is the standard deviation of a distribution over its mean, and for power-law distributions the CV is related to the power-law coefficient: a larger CV means the distribution has a heavier tail (i.e. there are hub regulators which have many target genes, or there are hub target genes which are directly affected by many regulators; see S1 Fig). We note that this algorithm does not pre-specify a partitioning of genes into regulators (with an outgoing edge) and non-regulators (without an outgoing edge); as formulated, any node can receive an outgoing edge, and so any gene in the network can act as a regulator.

## 2.3. Expression model

In order to enable reasonable comparisons with experimental data, we use an expression model with quantitative (rather than binary) measurements, and with dynamics subject to a non-linearity that enforces realistic physical constraints: gene expression is non-negative and saturates near a maximum value. Given a graph structure generated using the algorithm above, we assign parameters to each gene (node) and regulatory interaction (edge) in the graph. Each gene $i$ has two rate parameters: one for innate RNA production in the absence of regulators ($\alpha_i$), and another for the decay of existing cellular RNAs ($\ell_i$). Each regulatory relationship, between genes $j$ and $i$, has one parameter: a magnitude ($\beta_{ji}$), which describes the importance of the regulator for the expression of the target gene. We also enforce a constraint that interactions have a minimum strength ($|\beta_{ji}| \geq 1$). A full description of the strategy we use to sample these parameters for synthetic GRNs is in **Methods**.

Our expression model takes the form of a stochastic differential equation (SDE), and we produce expression values using forward simulation according to the Euler-Maruyama method (Fig 3A). For gene $i$ with regulators $j$ having expression $x_i$ (likewise $x_j$) at time $t$, the difference equation for expression $x'_i$ at time $t + \Delta t$ is given by

$$\frac{x'_i - x_i}{\Delta t} = \sigma\left(\alpha_i + \sum_j x_j \beta_{ji}\right) - \ell_i x_i + \mathcal{N}\left(0, s^2 \frac{x_i}{\Delta t}\right),$$

where the terms on the right hand side of the equation, in order, correspond to transcriptional synthesis, degradation, and noise. Unless stated otherwise, we set $s = 10^{-4}$ as the magnitude of noise, which serves to scale the intrinsic biological noise in the synthesis and degradation of RNAs (hence noise is also proportional to $x_i$). We let $\Delta t = 0.01$ be the step size, as in previous work [35], and take $\sigma(x)$ as the logistic sigmoid (expit) function $\sigma(x) = 1/(1 + e^{-x})$. In practice, we conduct forward simulation in vectorized form with an update rule:

$$x' = x + \Delta t \cdot \left(\sigma(\alpha + \beta^\top x) - \ell x\right) + \mathcal{N}\left(0, \Delta t \cdot s^2 \text{diag}(x)\right).$$

Throughout our experiments, we perform on the order of thousands of iterations and then check that the system of differential equations has reached an expression steady-state (**Methods**).

Our model can be used to quantify the effects of many types of perturbations. These include (1) gene knockouts (KOs), which we model by nullifying $x_j = 0$ (or equivalently, setting $\beta_{ji} = 0$ for all $i$); (2) gene knockdown or overexpression, which can be modeled by decreasing or increasing $\alpha_j$, increasing or decreasing $\ell_j$, or directly manipulating $x_j$ to a fixed value; (3) enhancer edits or transcriptional rewiring, modeled by changing specific $\beta_{ji}$; and (4) changes to expression noise, modeled by altering $s$, either globally or for specific genes. We

further note that similarly formulated perturbations with small magnitudes could also make appropriate models of the effects of molecular quantitative trait loci (QTLs). Here, we focus solely on gene knockouts, which we consider for the remainder of this work.

## 2.4. Perturbation studies

We conducted synthetic perturbation studies in 1,920 GRNs with $n$ = 2,000 genes; these GRNs were produced with a range of network generating parameters (**Methods**). For each GRN, we initialized gene expression values at zero and conducted a minimum of 5,000 iterations of forward simulation, later verifying that the dynamical system reached equilibrium and assessing its stability (Fig 3A and 3B, **Methods**). We then independently knocked out each gene in the network and let the system re-equilibrate after additional rounds of forward simulation (Fig 3B). We computed the effect of perturbing gene $j$ as the log-fold change in expression $x_i$ of all other genes $i$. We refer to this as the "perturbation effect" of gene $j$ on gene $i$. Mathematically, this is

$$\log_2 \text{FC}_{ji} = \log_2 \left( x_i | \text{do}(x_j = 0) \right) - \log_2 (x_i)$$

where $x_i | \text{do}(x_j = 0)$ denotes the expression of gene $i$ when gene $j$ has been knocked out. While most perturbation effects are small in all GRNs, with 86.6% of all effects below $|\log_2 \text{FC}|$ = 0.01, each network harbors a median 5,296 large effects on the order $|\log_2 \text{FC}|$ = 1 (Fig 3C). We also note substantial variability in the distribution of perturbation effects across networks (Fig 3C).

These effects are stratified by the distance between regulator and target (Fig 3D), with distance here being the length of the shortest path between genes along edges in the network. Across GRNs, the majority of direct regulators of a gene confer at least a modest effect on average (77.3% of genes at distance 1 have $|\log_2 \text{FC}| > 0.01$ when knocked out). Meanwhile, indirect effects of this magnitude also exist, but are less common (mean 21.5% of gene pairs not connected by an edge). However, since mediation is much more common than direct regulation, mediated effects contribute a substantial fraction of perturbation effects at all but the largest magnitudes. For example, 98.5% of effects at $|\log_2 \text{FC}| > 0.01$ across GRNs are mediated rather than direct (S2 Fig).

Since genes in the simulated GRNs belong to pre-defined groups, we further investigated the extent to which perturbation effects cluster within rather than across groups. On average, there is an enrichment of effects within groups—but as with the overall distribution, there is heterogeneity in the distributions of within- and across-group perturbation effects (Fig 3E). This heterogeneity is driven largely by the modularity term: as $w$ increases, the distributions of within- and across-group effects become further separated, even across networks with different numbers of groups (S3 Fig). This effect is based on changes in network architecture: since the strongest perturbation effects are from direct regulators, an increased affinity for within-group regulation (i.e., larger $w$) means that these effects should also come from members of the same group.

## 2.5. Impact of network properties

Next, we turned our attention to the relationship between properties of networks (as determined by network generating parameters) and their distributions of perturbation effects. As summaries of this distribution, and of the overall susceptibility of individual GRNs to perturbations, we compared the number of genes which are hub KOs and hub target genes in each of the 1,920 synthetic GRNs. We say a gene is a hub KO if it introduced a change of

A: Model gene expression

B: Compute gene knockout (KO) effects

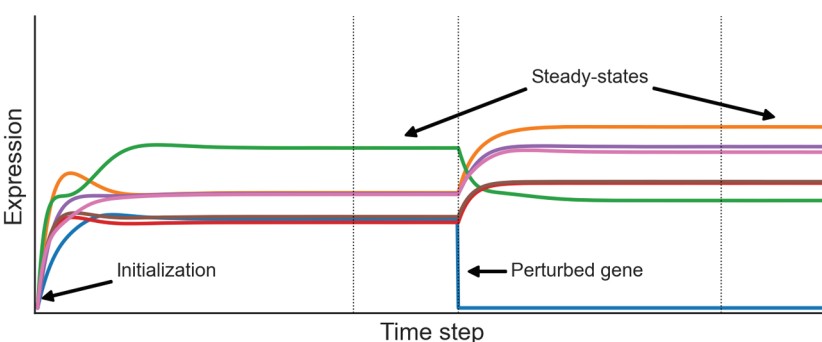

Stochastic differential equation (SDE) has terms for synthesis, degradation, and noise for gene $i$ at times $t$, $t+\Delta t$:

$$\frac{x_i^{[t+\Delta t]} - x_i^{[t]}}{\Delta t} = \sigma\left(\alpha_i + \sum_k \beta_{ki} x_k^{[t]}\right) - \ell_i x_i + s\sqrt{\frac{x_i^{[t]}}{\Delta t}} \mathcal{N}(0,1)$$

KO effect sizes are $\log_2$ fold-changes between expression steady-states:

$$\log_2 FC_{ji} = \log_2(x_i | do(x_j = 0)) - \log_2(x_i)$$

C: KO effects in 50 GRNs     D: KO effects by graph distance     E: KO effects by module

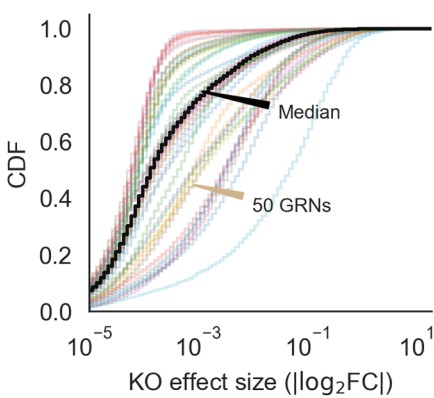
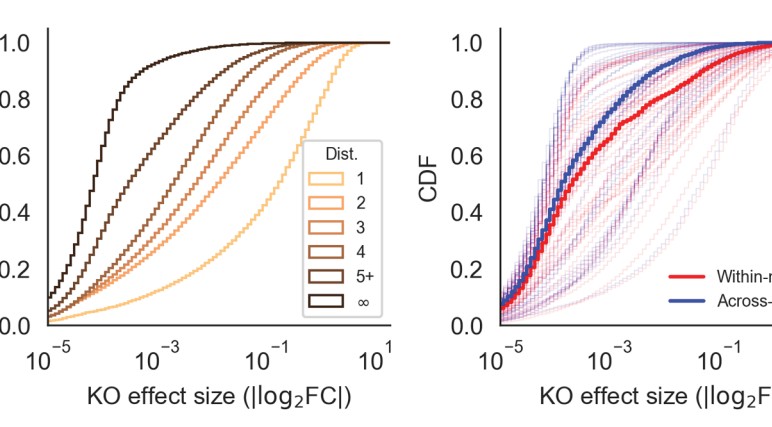

**Fig 3. Perturbations and their effects within networks**. **(A)** Overview of gene expression model and its parameters. Here, $\sigma$ is the logistic sigmoid $\sigma(x) = 1/(1 + e^{-x})$. **(B)** Example forward simulation of the dynamical systems model. Trace lines show genes, whose expression values are initialized at zero. The system eventually reaches a steady-state, and is then subject to perturbation (knockout of gene $j$, i.e. holding $x_j = 0$). Further forward simulation leads to a new steady-state, from which we can compute perturbation effects ($\log_2$ FC for other genes $i$). **(C)** Distribution of knockout (KO) effects (i.e., $\log_2$ fold-changes in expression $x_i$ of a focal gene $i$) in 50 example GRNs, along with the median distribution (black line). **(D)** KO effects as a function of network distance between two genes, and **(E)** within and across modules given by the generating algorithm. Note that the solid lines in **(D)** and **(E)** are the median distributions over the 50 example GRNs, split respectively by distances and modules.

$|\log_2 \text{FC}| > 0.1$ in at least 100 other genes when knocked out; analogously, we say a gene is a hub target gene if its expression was changed by $|\log_2 \text{FC}| > 0.1$ upon knockout of at least 100 other genes. Genes whose equilibrium expression was below the magnitude of noise were removed from these counts, as their expression could vary widely across conditions solely due to noise. We find that these statistics respond consistently to changes in network generating parameters (Fig 4), and that each parameter has the same direction of effect on the number of perturbation effects of this magnitude in the network (S4 Fig).

Graph sparsity has the greatest influence on the number of hub KOs and hub target genes in the GRNs (Fig 4A). More regulators per gene (large $1/p$) tends to translate to more perturbation effects overall, increasing the number of both hub knockouts and target genes. Notably,

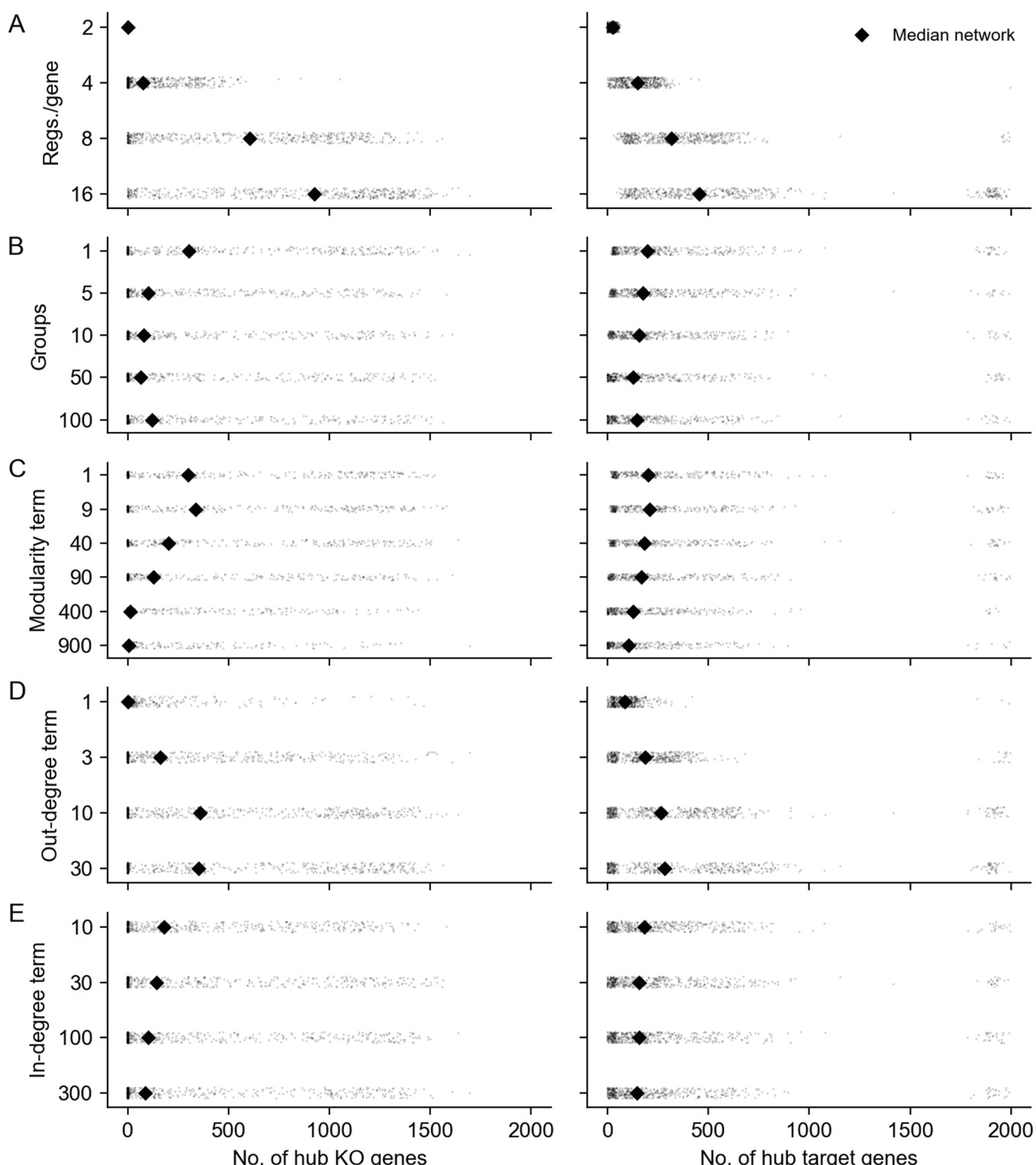

**Fig 4. Network properties influence their susceptibility to gene perturbations**. Counts of genes that are hub knockouts (**left**) and hub target genes (**right**) in each synthetic GRN, as a function of network generating parameters. Each panel shows all 1,920 GRNs as individual points, stratified by parameter values. Each distribution is annotated with its mean over GRNs (diamond points).

the effects on regulators and targets are not identical. In denser networks, the median number of hub KOs tends to be larger than the number of hub target genes. However, in a subset of dense networks, most genes in the network are identified as hub target genes. This is related to the absence of stable equilibrium dynamics in the low-noise limit of the gene expression model (S5 Fig), which suggests that as GRNs become more dense and genes are subject to regulation by larger fractions of the network, the system is less likely to be stable. Although this term has a large effect on perturbation effects, we find no obvious interaction between it and other terms in the generating algorithm (S6 Fig).

GRNs with fewer groups (small $k$) and higher modularity (large $w$) tend to have fewer hub KOs and hub target genes (Fig 4B and C). The modularity term monotonically increases resilience to perturbation; the group term monotonically decreases it, with the exception of $k = 1$. From the perspective of the network generating algorithm, $k = 1$ and $w = 1$ are identical; they are equivalent to the algorithm from Bollobás et al., 2003 [32] and correspond to the dissolution of modular structure with respect to the specified grouping. This is also equivalent to $k = 2000$, in which each gene in the network has its own group – this intuition is supported by the remaining trend in (Fig 4B). Meanwhile, in modular networks (large $w$), most edges are between members of the same group (Fig 4C). This might serve to confine the downstream effects of perturbations to members of the same group, effectively dampening the transcriptional impact of altering the function of master regulators.

When the out-degree distribution of GRNs has a heavier tail (small $\delta_{out}$), there tend to be many fewer hub knockout genes (Fig 4D). This relationship is non-linear, and in the most extreme case ($\delta_{out} = 1$) there are only 89 hub KOs on average (median 1 hub KO) in the GRN. This effect is a consequence of preferential attachment: as more edges are drawn from master regulators, outgoing regulatory effects also concentrate there. Counterintuitively, this parameter exerts influence over the number of hub target genes in the network as well, and in the same direction. When effects are concentrated among a few key regulators, it may simply be less feasible for any gene to be affected by many knockouts (since there are fewer genes that have many knockout effects at all). As with the sparsity term, we do not see obvious interactions between this term and others in the generating algorithm (S7 Fig). Meanwhile, when the in-degree distribution of GRNs has a less heavy tail (large $\delta_{in}$), there are modestly more hub target genes and hub KOs (Fig 4E). The source of this trend is difficult to intuit, but the effect is very weak.

Looking across parameters, these results reflect a wide range of variation in the susceptibility of GRNs to perturbations as a function of their structural properties. While there is substantial overlap in the distributions of hub KO and target genes across network generating parameters, we find that all parameters except the in-degree term have statistically significant effects on both quantities ($p < 0.001$ for all tests, S8 Fig—full results for outgoing effects in S1 Table and for incoming effects in S2 Table). We estimate these effects with a multiple regression on the logit-transformed fraction of genes in each GRN which are hub KOs or hub target genes (**Methods**). In total, the network generating parameters explain just under half the variance in the fraction of the GRN which is either a hub KO (model $r^2 = 0.59$) or hub target gene (model $r^2 = 0.46$)—the remaining variance is attributable to randomness in the generating algorithm and in the expression model parameters (**Methods**). Moreover, there is also a noteworthy thematic consistency across parameters. The direction of protective effect from perturbation was consistently that of biological realism: GRNs with fewer hub KOs and target genes tended to be sparser, more modular, and had a heavier tailed out- but not in-degree distribution.

## 2.6. Comparing with experimental data

With this intuition about network properties in hand, we now return to real experimental data. Given that synthetic GRNs with quantifiably different structures produce qualitatively different distributions of knockout effects, we next sought to ask whether any of them were also similar to real data. For this, we used the subset of perturbations that correspond one-to-one with gene expression readout in a recent Perturb-seq study [9]. We used data from this study in particular for two reasons: its scale (thousands of genes measured), and its unbiased selection of genes to perturb. These features contrast with other work focused on specific pathways [6] or genes expected to have large transcriptional effects when perturbed (e.g., "essential" gene experiments from the same study [9]). These features are unique in the present literature, and we find them to be critical for understanding global features of regulatory networks at the resolution we seek to model.

We compared the real and synthetic data using their cumulative distributions of perturbation effects, computed both from the perspective of genes as regulators and as targets of regulation (Fig 5A and 5B, **Methods**). Since these data have different numbers of genes that are not lowly expressed, we normalized the number of incoming and outgoing effects to the size of the network (S9 Fig, **Methods**). In the Perturb-seq data, we find noticeable qualitative differences between the distribution of incoming and outgoing effects. These differences are consistent with the notion that GRNs should have master regulators but not master target genes. In the synthetic data, we find substantial diversity in both distributions across GRNs, including many patterns that seem wholly incompatible with those observed in experiments.

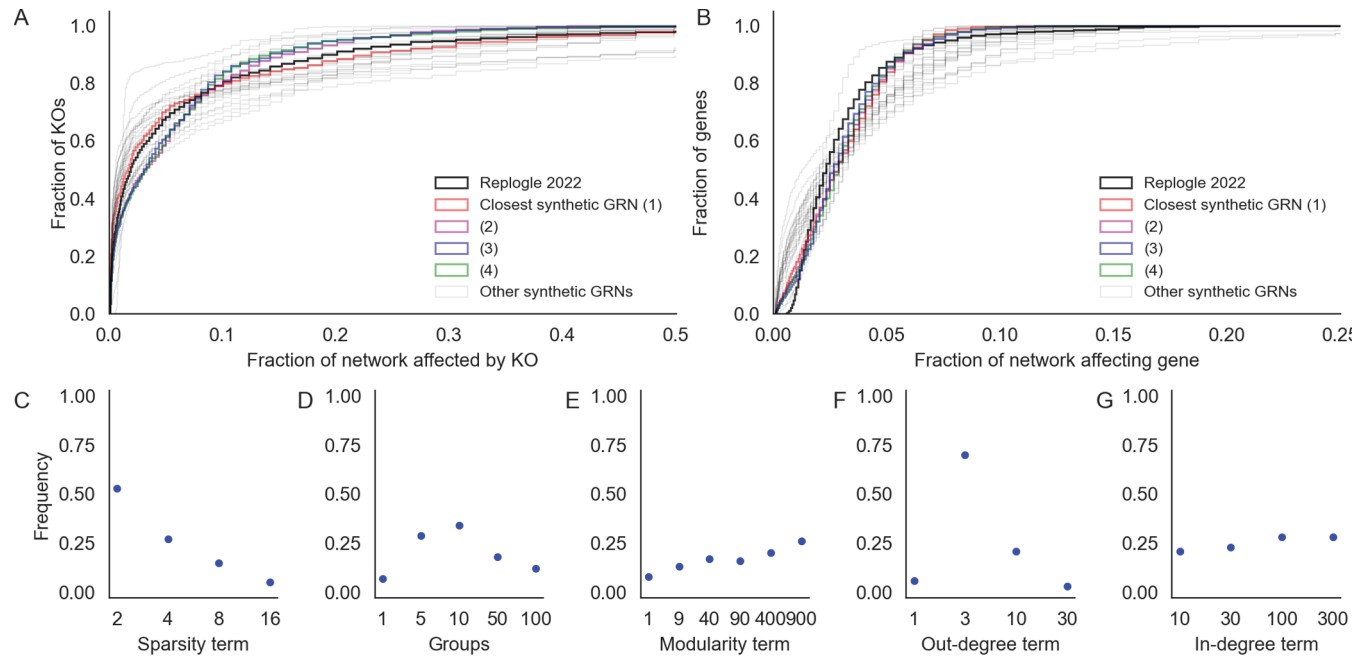

**Fig 5. Comparing with genome-wide Perturb-seq**. Fraction of GRN that **(A)** affects each gene when perturbed (outgoing effects) or **(B)** is affected by other perturbations (incoming effects). In synthetic data, perturbation effects are thresholded at the top 3% of absolute log-fold change values, matching the proportion of pairwise tests from the Perturb-seq data with FDR-corrected Anderson-Darling $p < 0.05$. Highlighted in color are the four GRNs that best match the Perturb-seq data. **(C-G)** Distribution of network generating parameters for the 100 GRNs that are best matched to Perturb-seq data (by Kolmogorov-Smirnov $p$-value rank for both distributions in A and B).

Meanwhile, some GRNs seem well matched to the Perturb-seq results: the distributions closest to the data are highlighted in color in Fig 5A and 5B, and correspond to those having the smallest Kolmogorov-Smirnov test statistics when compared with the data distributions (**Methods**).

Although the focus of our work is not wholesale network inference, we do observe a coherent set of properties among well-matched networks (Fig 5C–G). Specifically, they share a relatively small number of regulators per gene (two or four, rather than 16); they have a small number of groups (five to 10 rather than one or 100); they are highly modular (large $w$); and they have a heavy tail in the distribution of out-degree but not in-degree ($\delta_{out}$ near three but $\delta_{in}$ on the order of 100). Consistent with previous results, we find these parameter sets to be within a range that matches our motivating intuition about the structural properties of GRNs. We further do not find these properties to have obvious pairwise interactions that affect concordance to data (S10 Fig). Moreover, these results are also consistent across rank-based thresholds for the matching (S11 Fig) and across resampled parameters for the expression model, given the same causal network underlying the GRN (S12 Fig).

It is worth noting that while some of the networks seem well matched to real data by eye, they are all statistically distinguishable from the observed distributions of incoming and outgoing effects (S1 Data). This is expected behavior since the networks we generate are entirely synthetic, and leverage no specific prior biological information about the structure of the human genome or statistical properties of this specific dataset. We suspect that a more flexible specification of group size, number, and affinity (i.e., altering the $w$ parameter) could improve concordance with the distribution of outgoing effects (Fig 5A), and that modeling the detection power of experimental assays could improve concordance with the distribution of incoming effects (Fig 5B). This latter point is especially important as baseline expression is a stronger determinant of the number of detected incoming effects in real data compared to simulations (which may reflect power limitations in the Perturb-seq data; S13 Fig), even as network properties also shift the tail of the distribution of $|\log_2 FC|$ (S14 Fig and Fig 3A). Despite these limitations, we find that all parameters of our algorithm are necessary to match the observed distribution from data. Notably, it is not sufficient to let the group structure be an emergent property of the network (as in Bollobás et al. 2003), which corresponds to $k = 1$ (or $w = 1$) in our model. These networks do not match real data as well as those with enforced group structure (Fig 5D and 5E).

## 2.7. Challenges and opportunities for inference

Finally, we highlight the utility of our simulation approach by considering the value of different data sources for inference tasks. For this, we conducted further analysis using an example synthetic GRN whose patterns of knockout effects were well-matched with Perturb-seq data. Specifically, we focused on the recovery of edges, edge weights, and group structure using interventional data (e.g., perturbation effects) and observational data (e.g., coexpression). We made use of perturbation effects as previously described, and further computed pairwise gene coexpression values using additional rounds of forward simulation from the expression model at steady state to approximate the naturally occurring variation across cells (**Methods**).

**2.7.1. Discovering pairwise relationships.** Several computational and experimental approaches have been used to estimate pairwise causal relationships between genes, with the ultimate goal of wholesale inference of gene regulatory networks [10,11]. These data and methods are broad in scope, and range from estimating networks using natural variation in gene expression values from bulk tissue [12,23] to fitting complex machine learning models

on data from single-cell perturbation experiments [7,15,36]. Here, we consider two descriptive pairwise summary statistics at the gene level—gene coexpression across cells and perturbation effects across gene knockouts—and their connections to edges in a simulated GRN.

In the synthetic data, we find that the distributions of pairwise gene coexpression values and knockout effects span multiple orders of magnitude (Fig 6). However, where the distribution of knockout effects differs dramatically between gene pairs that share an edge and those that do not, the distributions of coexpression values have substantial overlap (Fig 6A and B). This difference in distribution reflects what each statistic tends to measure. Gene perturbation effects tend to flow through the network along edges and are therefore highly related to the network distance between genes (Fig 3D). This includes whether or not two genes share a direct regulatory relationship in the form of an edge (S15 Fig). Meanwhile, strong coexpression is more often due to co-regulation than to direct causal relationships between genes (S15 Fig), making it difficult in practice to uncover edges with this data.

For gene pairs where there is direct regulation, we see that both knockout effects (Fig 6C) and coexpression (Fig 6D) have weak correlation with the strength of known regulatory relationships. This reflects that both statistics are imperfect measures of regulatory importance: they are both affected by differences in regulatory architecture across genes (e.g., the number of regulators or the intensity of transcriptional buffering). We further see this when comparing coexpression and knockout effects between pairs of genes. Across all pairs of genes, these two statistics are uncorrelated—but the two are highly correlated among pairs of genes that share an edge (Fig 6E). In this way, both perturbation effects and coexpression contain similar information about edges in the GRN—but coexpression also measures non-causal relationships between genes, like coregulation, and is therefore systematically uncorrelated with perturbation effects even in real data (S16 Fig).

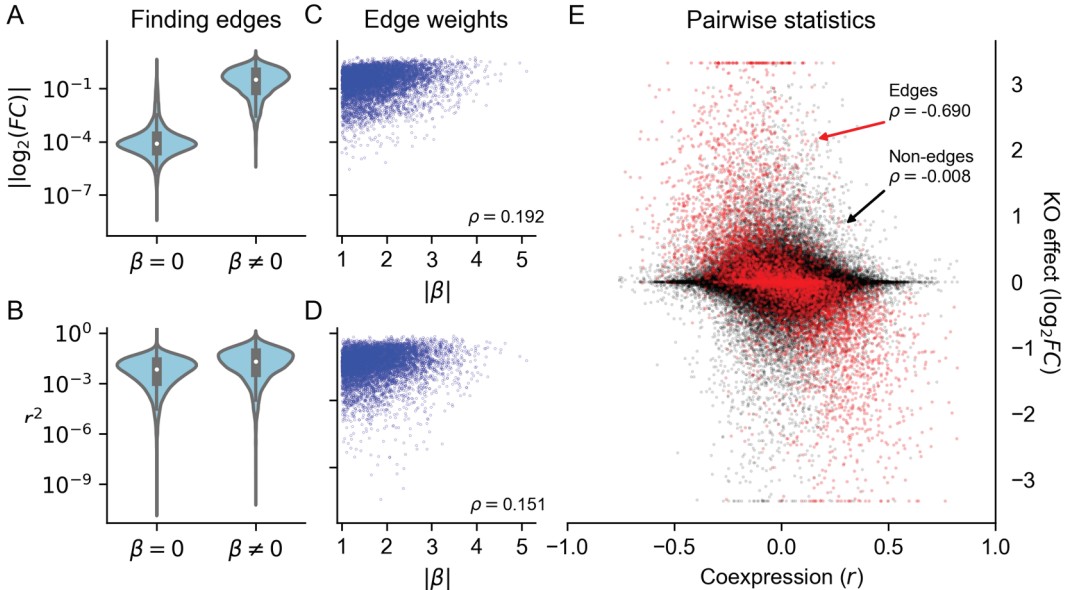

**Fig 6. Perturbations more reliably measure fine-scale network structure than coexpression**. (**A**) Distribution of perturbation effects and (**B**) coexpression values between pairs of genes that do or do not share an edge in a realistic synthetic GRN (both pairs of distributions have statistically distinguishable means; $p < 10^{-16}$, two-sample $t$-test). (**C**) Rank correlation of perturbation effect sizes and (**D**) coexpression magnitude with edge weights. (**E**) Rank correlation between coexpression and perturbation effects (the $y$-axis is clipped at values corresponding to tenfold change).

We might therefore expect that network features associated with increased coregulation (like modularity and the presence of transcriptional master regulators) will widen the gap in performance between coexpression and perturbation effects. Indeed, across networks in our study, we find that sparsity, modularity, and the existence of hub regulators (small $\delta_{out}$) tended to diminish the enrichment of true edges among pairs of genes with high coexpression (S17 Fig). The opposite is true for perturbation effects, which are better predictors of edges in these same networks—however, it is worth noting that for finding edges, perturbation effects outperformed coexpression summary statistics in every network in our study (S17 Fig).

Taken together, these results underscore the importance of perturbation data for inferring network edges, and may suggest limitations in the use of coexpression networks. However, neither form of data is a panacea, and care is warranted in the analysis of real experimental data and in the development of structure learning algorithms. In particular, we anticipate that incorporating prior information on structural properties of regulatory networks will improve GRN inference techniques. As a simple example, we show that biasing the rank ordering of perturbation effect p-values from the Replogle et al. dataset (to encourage edges to be drawn out from putative master regulators) improves correspondence with pairwise gene links from external sources on protein-protein and ChIP-seq interactions (**Methods**; S18 Fig).

**2.7.2. Discovering group structure.**   Recent work has also attempted to identify trait-relevant sets of genes that act through coordinated effects in a particular cell type. These groups are sometimes called programs, and it is common to use dimensionality reduction techniques like singular value decomposition (SVD) or non-negative matrix factorization (NMF) on single-cell expression values to identify groups [8,37]. In our example synthetic GRN and in the Perturb-seq data, we used a variant of this approach based on truncated SVD (TSVD) to assign genes to programs. As input, we used the set of 75,328 unperturbed cells from real data [9] and downsamples of the entire experiment to the same number of cells; for the synthetic data, we simulated the same number of cells from baseline or baseline and perturbed conditions, mimicking the composition of the real experiments. From these data, we computed the first 200 singular vectors of the expression data, using each vector to define a "program" of 200 genes with the largest loadings (**Methods**).

Here, we assess the extent to which these programs and their constituent singular vectors replicate across experiments from perturbed and unperturbed conditions. We used canonical correlation analysis (CCA) to assess the similarity of the singular vectors. This technique seeks to find rotations $a,b$ of inputs $x,y$ such that the correlation between $a^\top x$ and $b^\top y$ is maximized. The transformed inputs are called canonical variables, and we report their correlations when the inputs $x,y$ are gene expression vectors from different experimental conditions (Fig 7A), reduced into 200 dimensions of input. In the synthetic GRN, the canonical correlation steadily declines. Notably, the magnitude of this correlation is similar when perturbation data are compared to a replicate or to unperturbed data. Even though this correlation is modest by the 100th set of canonical variables, this trend suggests consistency between lower-dimensional representations of expression data regardless of cell state (perturbed or unperturbed).

However, at this sample size (75,238 cells) for the synthetic GRN, there is a difference in the reproducibility of individual programs between data sources (perturbed or unperturbed; Fig 7B). To show this, we compared programs computed from one perturbation experiment (the "reference") to programs from a replicate perturbation experiment and to programs from unperturbed cells. For each program in the reference, we report the maximum number of genes that overlapped with any other program computed from each of the other data

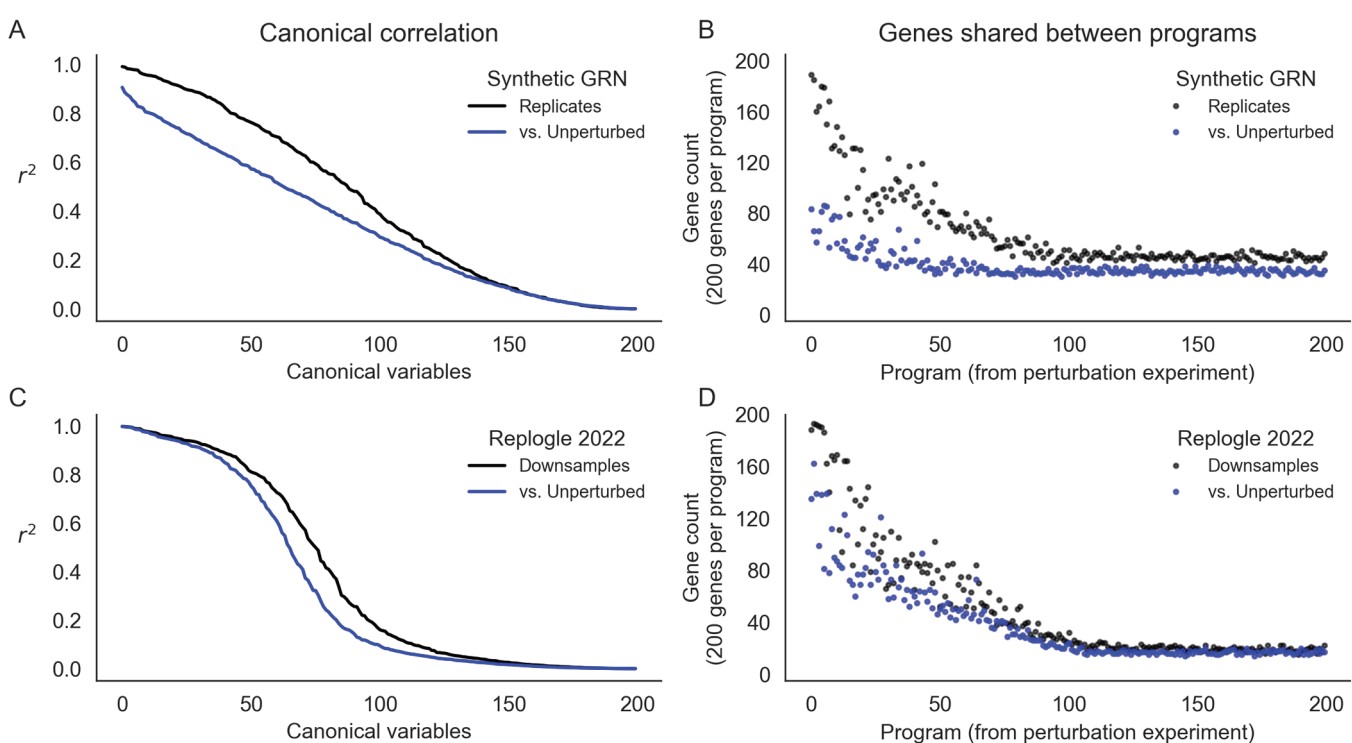

**Fig 7. Learned representations are similar between control and intervened-upon cells**. Concordance between low-rank representations of single cells in a simulated GRN (top row) and downsamples of experimental data (Replogle et al., 2022). **(A)** Correlation between the first 200 canonical variables (linear combinations of singular vectors) between samples of 75,328 baseline or baseline+perturbed cells from a synthetic GRN. **(B)** Overlap in gene programs inferred from singular value decomposition of single cell expression data. Programs are defined using singular vectors of gene expression from perturbed cells (x-axis), and intersected with programs analogously defined from baseline and additional perturbed cells. **(C)** Canonical correlation of control cells and two downsamples of the entire Perturb-seq experiment (Replogle et al., 2022). **(D)** Overlap in gene programs computed from control and downsamples of experimental Perturb-seq data.

sources. The first few programs (corresponding to the first few singular vectors) are highly reproducible in the replicate perturbation data. This overlap steadily declines to zero effective overlap after the first ~100 programs. Despite similar canonical correlation, the unperturbed data do not recapitulate the same programs—there is modest overlap with the first few programs from the perturbation data, and this overlap decays very quickly after the first ~20 programs. When comparing these programs to the ground truth groups of this network ($k = 10$), we find that nearly all true groups are at least modestly well represented by the top 50 programs from both data sources. However, the programs from the perturbation data better represent the true groups than those from the unperturbed data (S19 Fig).

We find similar results in the genome-wide Perturb-seq data. Here, however, the first few canonical variables are highly correlated, and the canonical correlation drops precipitously between the 50th and 100th canonical variables (Fig 7C). We also find that programs computed from two downsamples of the entire experiment are about as reproducible as those from the simulated data, but are slightly more similar to the programs from control cells (Fig 7D). While this may reflect some aspect of GRN structure, it is also related to the number of cells in the input data and the magnitude of intrinsic gene expression noise. Both tend to reduce canonical correlations and the reproducibility of gene programs. In the real data, lowering the number of cells input to TSVD lowers both canonical correlation and program similarity compared to the entire experiment; relatedly, we find that 75,328 control

cells exhibit comparable performance to 30,000 perturbed cells at recovering representations from the full data (S20 Fig). In the synthetic networks, we find that altering the level of global transcriptional noise ($s$) alters the concordance between replicates and between perturbed and unperturbed cell state. At high levels of noise, there is little practical difference between programs derived from perturbed or unperturbed cells, but with low levels of noise, the programs from perturbation data are markedly different (S21 Fig). For presentation in Fig 7A and B, we chose a level of noise ($s = 0.3$) that recapitulated the qualitative behavior of the real data.

Taken together, these findings seem to suggest that the leading variance components of single-cell gene expression data will be similar across perturbed and unperturbed conditions, unless the magnitude of perturbation effects is greater than the level of intrinsic transcriptional noise. Given this, we suspect that dimensionality reduction techniques will produce concordant representations of both perturbed and unperturbed data and that this similarity can propagate into gene sets derived from these representations. This begs a key line of questions for future work: where, and how, do molecular perturbations add value in uncovering the sets of genes that are collectively important for cell-type and disease-specific processes? In light of the number of cells required for reliable inference at this scale, we anticipate that large atlas-style cell reference data (e.g., the Human Cell Atlas and similar resources [38–40]) should provide promising opportunities to reveal global aspects of network structure.

## 3. Discussion

In this work, we have presented a new model to simulate gene regulatory networks, with particular emphasis on generating networks with realistic structural properties. We note that this algorithm may be of interest in contexts outside gene regulation – namely, in studies of scale-free networks with group-like structure. We also anticipate that our technique to simulate gene expression from arbitrary networks may be useful for model development and benchmarking, or in other studies where network structures are known or may be hypothesized.

Here, we have highlighted the utility of our approach with simulations to develop intuition about key properties of GRNs, particularly in the context of molecular perturbations and coexpression data. While our study design draws inspiration from recent works using Perturb-seq, we also acknowledge that our model makes simplifying assumptions. In focusing on the equilibrium dynamics of cells of one type, we have ignored developmental trajectories and cell-type heterogeneity within tissues, both of which modify our assumptions about regulatory network structure. For the sake of computational efficiency in quantifying the expression of thousands of genes, we have also eschewed detailed models of the biological synthesis and experimental measurement of cellular RNAs (e.g., models that use the Hill equation [35] or account for interactions between regulators [25]). For the same reason, our model also does not account for post-transcriptional activity (e.g., models of translation and protein-DNA and protein-protein kinematics), although the regulatory parameters of our expression model (i.e., beta coefficients within the sigmoid) may have some approximate relationship with parameters in a more fully specified biophysical model of expression regulation.

In future work where it is critical to model protein kinematics or to match distributions of RNA count data from experiments, we encourage modeling these complexities. We note that as in prior work [25,35], such models of protein translation, RNA observation, or other kinematic activity could be layered on top of our baseline expression model to produce data with more desirable summary statistics. Moreover, our modeling approach is flexible enough to

accommodate an entirely separate expression model which could be specified given an underlying causal graph. To facilitate this type of extension to our model, we have provided our analysis code on github (https://github.com/maguirre1/grn-paper), with separate implementations of our graph generating algorithm and expression model.

Independently, our results suggest that the space of realistic network structures may be quite limited, and that it may be useful to consider this prior information in various inference settings. While the approach outlined in this work is not optimized for inference, the algorithms we describe are generative, which means that they could be used directly in applications for simulation-based inference. Although we used experimental data from K562 cells in this study, we anticipate that the high-level structural properties of GRNs will generalize across different cell types. Moreover, we observed through simulations that hallmark properties of GRNs tend to confer resilience to perturbations across multiple measures, reducing the number of sensitive target genes and large-effect master regulators. We do not suspect this to be an incidental finding in light of the selection to which GRNs are subject over evolutionary time, and we suspect that considering this type of constraint may be insightful for future work.

Looking forward, we anticipate that broad observational studies of diverse cell types and deep interventional studies of specific cell lines will both be useful in disentangling the basis of complex traits in regulatory networks. However, a key question remains in determining how best to leverage these data types towards a unifying understanding of cell biology. We suggest that a scaffolded approach to this problem may be useful. Where the scale of cell atlases presents a unique opportunity to learn transferable representations of cells across developmental states and tissues, perhaps including the discovery of cellular programs, these data are limited in their ability to resolve interactions between single genes. This is where perturbation data—however limited to specific cell types—retain critical value. Even as existing network inference algorithms experience computational challenges in genome-scale applications, the modularity of GRNs suggests that piecewise inference strategies may be viable until these challenges are resolved. As efforts like these enhance our mechanistic understanding of biological networks, we hope that our work serves to provide general intuition on their salient structural properties. We are optimistic that understanding these principles will be useful in addressing an array of challenges and highlighting future opportunities in functional genomics.

## 4. Materials and Methods

### 4.1. Graph generating parameters

**4.1.1. Sampling.** A full description of the graph generating algorithm can be found in the main text, with the exact procedure given by Algorithm 1. Here, we provide additional intuition on its generating parameters, and detail our scheme for sampling them.

In motivating our study, we highlight several key properties of gene regulatory networks: briefly, these are sparsity, modular groups, and asymmetric power-law degree distributions. In Fig 2 we show that these properties are individually tuned by parameters of our generating algorithm. When generating synthetic networks, we sample values for each parameter across one to three orders of magnitude. To cover these ranges, the values are spaced geometrically, and the extrema are chosen to overlap values which we believe are consistent with biological intuition for a network of $n = 2,000$ genes.

- Sparsity term $p$: $\left\{ \frac{1}{2}, \frac{1}{4}, \frac{1}{8}, \frac{1}{16} \right\}$.
- Number of groups $k$: $\{1, 5, 10, 50, 100\}$.

- Modularity term $w$: $\{1, 9, 40, 90, 400, 900\}$.
- In-degree uniformity term $\delta_{in}$: $\{10, 30, 100, 300\}$.
- Out-degree uniformity term $\delta_{out}$: $\{1, 3, 10, 30\}$.

The sparsity term $p$ is sampled so that the average number of regulators per gene spans from low single-digits to low double-digits. The number of groups is sampled from $k = 100$, which corresponds to a rough lower limit on the size of groups (20 genes), to $k = 1$, which corresponds to the dissolution of group structure and is equivalent to the algorithm from Bollobás et al., 2003 [32]. The within-group affinity term $w$, which controls modularity, is sampled in a similar way: $w = 1$ also corresponds to the dissolution of group structure, again giving the algorithm from Bollobás et al., 2003, and $w = 900$ gives an upper limit on modularity with respect to groups $k$. The in- and out-degree uniformity terms $\delta_{in}$, $\delta_{out}$ are both sampled across orders of magnitude. The bounds for the in-degree term are larger, corresponding to the assumption that the in-degree distribution should be less dispersed (i.e., have fewer hubs) than the out-degree distribution, but the range of values is intentionally overlapping.

To produce the set of 1,920 GRNs used in the study, we simulated one network with every possible combination of parameters listed above: this totals $4 \times 5 \times 6 \times 4 \times 4 = 1,920$ networks.

**4.1.2. Relationship to perturbation effects.** We performed a regression analysis to estimate the effect of each graph generating parameter on the distribution of perturbation effects in the synthetic GRNs. Specifically, we regressed the logit-transformed fraction of genes in each GRN that are hub regulators or hub targets according to the following equation:

$$\mathrm{logit}(p_{\mathrm{genes}}) \sim 1 + (1/p) + k' + w + \delta_{in} + \delta_{out},$$

where $1/p$ is the inverse of the sparsity term, $k'$ is a transformed number of groups (GRNs with $k = 1$ group are treated as GRNs with $k = n = 2000$ groups; see Fig 4), and $w$, $\delta_{in}$, and $\delta_{out}$ are as described above. The dependent variable of the regression, $p_{genes}$ is either the fraction of genes in the GRN which are hub regulators or hub targets. These quantities are analyzed separately. Full results for outgoing effects are in S1 Table, and full results for incoming effects are in S2 Table.

## 4.2. Expression simulation

**4.2.1. Parameter selection.** An overview of our gene expression model can be found in the main text. Here, we describe the sampling strategy for the parameters of the model and provide additional information on their interpretation. Recall that the expression, $x_i$, of gene $i$ is influenced by the following variables:

- the baseline transcription rate, $\alpha_i$,
- the degradation rate of RNAs, $\ell_i$,
- effects from regulating genes, $\beta_{ji}$,
- expression noise, with magnitude $s$.

Note that $\alpha$ and $\ell$ are properties of genes (nodes); $\beta$ is a property of regulatory interactions (edges); and $s$ is a global parameter for the entire network. During forward simulation from the discretized stochastic differential equation, we take steps of size $\Delta t = 0.01$ as in prior work [35], and update expression values from $x$ (at time $t$) to $x'$ (at time $t + \Delta t$) according to the

following:

$$x' = x + \Delta t \cdot (\sigma(\alpha + \beta x) - \ell x) + s\sqrt{\Delta t \cdot x}\mathcal{N}(0, I).$$

In the deterministic limit, this results in an equation satisfied by any potential steady-state

$$x^* = \sigma(\alpha + \beta x^*)/\ell,$$

where $\sigma(x) = \frac{1}{1+e^{-x}}$ is the logistic sigmoid. When setting up the model given a graph structure from our generating algorithm, we simulate expression parameters according to the following scheme:

- $\sigma(\alpha_i) \overset{\text{iid}}{\sim} \text{Beta}(2, 8)$, under the assumption that genes have low but non-zero expression at baseline, in the absence of regulation—i.e., $\sigma(\alpha_i)$ is small. Here, $\sigma(x) = \frac{1}{1+e^{-x}}$ is again the logistic sigmoid (expit) function.
- $\ell_i \overset{\text{iid}}{\sim} \text{Beta}(8, 2)$, under the assumption that the maximum expression of each gene, $1/\ell_i$, tends to be of a similar order of magnitude (close to one), but can vary. To prevent steady-state gene expression from being excessively large (by having small $\ell_i$), we hard clip $\ell_i$ to be at least as large as $e^{-\alpha_i}$.
- $\beta_{ji} \overset{\text{iid}}{\sim} (2p_j - 1) \cdot (1 + \text{Half-Normal}(0, 1))$, under the assumptions that regulatory interactions have a minimum strength ($|\beta_{ji}| \geq 1$). Here, $p_j \overset{\text{iid}}{\sim} \text{Bernoulli}(p = 0.8)$ is the probability that a regulator $j$ acts as an activator rather than a repressor.
- $s = 10^{-4}$, fixed across all genes in the networks. This value is chosen to be as large as possible without limiting the detection of very small KO effects. At this level of noise, we can reliably detect $\log_2$ fold-changes down to the order of $10^{-4}$ (Fig 3D).

**4.2.2. Forward simulation.** Once parameters of the expression model are chosen for a specific GRN, we initialize the expression of each gene $x_i = 0$ and conduct forward simulation according to the update rule given in the previous section, which is also described in the main text.

When performing forward simulations, we initialize all genes in the network to have zero expression. We then perform $b = 5,000$ iterations of forward simulation as a "burn-in". After burn-in, we check whether the system of equations has converged to a steady-state by measuring differences in the time averaged mean after the burn-in. Specifically, we compute the maximum absolute $\log_2$ fold-change of non-lowly expressed genes in the network

$$\max_{g, \bar{x}_{g,i} > s} \log_2\left(\frac{\bar{x}_{g,i}}{\bar{x}_{g,i-h}}\right)$$

where $g$ indexes genes whose running mean expression $\bar{x}_{g,i}$ at the current iteration $i$ is above the noise magnitude $s$, and $h$ is the step size to check for convergence. Mathematically, the running mean in the numerator is

$$\bar{x}_{g,i} = \frac{1}{i-b} \sum_{t=b}^{i} x_{g,t}$$

where $x_{g,t}$ is the expression of gene $g$ at iteration $t$. The denominator $\bar{x}_{g,i-h}$ is analogously the running mean expression of gene $g$ the last time we checked for convergence,

$$\bar{x}_{g,i-h} = \tfrac{1}{i-h-b} \sum\nolimits_{t=b}^{i-h} x_{g,t}.$$

If this maximum log fold-change is below $10^{-3}$, we say the system has converged, and take the vector $\bar{x}_i$ as the steady-state expression of all genes in the network. We perform this check every $h = 1{,}000$ iterations, up to a maximum $t_{max} = 20{,}000$ iterations. We take the vector $\bar{x}_{t_{max}}$ as an approximate equilibrium state if the system did not fully converge.

We further assess the stability of the steady-state of each GRN by performing a linear stability analysis of the expression model in the limit $s \to 0$. In this limit, the expression model takes the form of an ordinary differential equation (ODE). The stability of an equilibrium point $\bar{x}$ of this ODE can be assessed using the eigenvalues of the Jacobian matrix $J$ evaluated at $\bar{x}$—if all of the eigenvalues have a negative real part, the system is said to be stable [41]. Here, we have

$$J = \left\{ \frac{\partial f(x_i)}{\partial x_j} \right\}_{ij}$$

where the $(i,j)$th entries correspond to the partial derivative of the deterministic part of the expression function $f(x_i)$ of gene $i$, with respect to the expression $x_j$ of gene $j$. For our model,

$$\begin{aligned}
\frac{\partial f(x_i)}{\partial x_j} &= \frac{\partial}{\partial x_j} \left( \sigma(\alpha_i + \sum\nolimits_k \beta_{ki} x_k) - \ell_i x_i \right) \\
&= \beta_{ji} \sigma(\alpha_i + \sum\nolimits_k \beta_{ki} x_k) \left[ 1 - \sigma(\alpha_i + \sum\nolimits_k \beta_{ki} x_k) \right] - \mathbb{1}(i = j)\ell_i,
\end{aligned}$$

where the first term is zero for $\beta_{ji} = 0$ and the second term is zero for $j \neq i$.

**4.2.3. Perturbation experiments.**   For each synthetic GRN in this study, we perform a systematic assessment of gene-level perturbation effects. We start with baseline steady-state expression values of an instantiated GRN, with edges drawn according to the generating algorithm and expression parameters chosen as described above. Then, separately for each gene $j$, we perform a knockout by setting $\beta_{ji} = 0$ for all other genes $i$—that is, we nullify its outgoing effects. This perturbs the equilibrium dynamics of the expression SDE, and we conduct additional rounds of forward simulation using the modified parameters until a new expression steady-state is reached. We perform the same procedure with burn-in and convergence checks as in the previous section.

We summarize the effect that perturbing gene $j$ has on gene $i$ using a log fold-change in expression values:

$$\log_2 \mathrm{FC}_{ji} = \log_2\left(x_i | \mathrm{do}(x_j = 0)\right) - \log_2(x_i)$$

where $x_i$ is the steady-state expression of gene $i$ under baseline conditions, and $x_i | \mathrm{do}(x_j = 0)$ is its steady-state expression when gene $j$ has been perturbed (both computed as described above).

**4.2.4. Baseline coexpression.**   Since gene coexpression is also commonly used to describe pairwise relationships between genes, we further use the expression SDE to compute the gene-level correlations at steady-state in the synthetic GRNs. Under baseline conditions, we

perform $t = 10,000$ additional forward time steps, from which we sample $m = 10,000$ "baseline cells" by taking the gene expression value at every step. The noise inherent to the model ($s = 10^{-4}$) produces sufficient variability in this cell population to compute gene-level correlations. We measure the coexpression of genes all $i$ and $j$ (not filtering out lowly expressed genes) using the Pearson correlation between $x_i$ and $x_j$ across cells.

### 4.3. Perturb-seq data

**4.3.1. Data processing.** To motivate aspects of our work, and to assess our simulations in the context of experimental data, we made use of summary statistics from a recent genome-scale Perturb-seq study [9]. Specifically, we used pairwise FDR-corrected Anderson-Darling p-values (from the supplemental file, ``anderson-darling p-values, BH-corrected.csv.gz``) as a measure of the expression response to single-gene perturbations. Throughout this work, we used a single large subset of these data corresponding to the set of genes whose expression was subject to both experimental perturbation and measurement in response. We matched perturbations to target genes using the provided ENSEMBL gene IDs, subsetting to perturbations which targeted any primary transcript. In (rare) cases where there was more than one such perturbation, we used the perturbation which induced a statistically significant change in the expression of the target transcript. We note that target genes with expression levels below 0.25 UMI per cell were not included in this file, which further limited the genes included in our analysis. We performed a similar post-processing step when analyzing results from the synthetic networks, limiting analysis to genes whose steady-state expression was above the level of intrinsic noise (i.e., $x_i > s$; values in S9 Fig). This resulted in an analysis set of 5,247 genes that were both perturbed and measured in the study.

**4.3.2. Comparing with simulations.** We compared the distribution of perturbation effects (incoming and outgoing) at the gene level when assessing similarities between the real and simulated networks. For this, we thresholded pairwise effects from the experimental data at FDR-corrected Anderson-Darling $p < 0.05$, saying that effects at this significance level constitute biologically meaningful effects, and others do not. At this threshold, we find that 3.16% of pairwise effects are called significant. For a given gene $i$, we then computed two values: the fraction of the network that is affected when $i$ is perturbed (i.e., the fraction of genes $j$ for which $p_{ij} < 0.05$), and the fraction of the network that affects $i$ when perturbed (i.e., the fraction of genes $j$ for which $p_{ji} < 0.05$).

We then compared these distributions to analogous quantities derived from the synthetic GRNs. Since the experimental data are affected by imperfect statistical power, we set the discovery rate to be equal across all synthetic GRNs, doing so by taking the top 3.16% of pairwise perturbation effects (i.e., $|\log_2 FC|_{ji} > k$, where $k$ variesl=; values in S14 Fig) as "statistically significant". For each gene $i$ in a synthetic GRN, we computed the fraction of the network which is affected when $i$ is perturbed (i.e., the fraction of genes $j$ for which $|\log_2 FC|_{ij} > k$), and the fraction of the network which affects $i$ when perturbed (i.e., the fraction of genes $j$ for which $|\log_2 FC|_{ji} > k$). Note that in each GRN in Fig 5, we remove lowly-expressed genes, with baseline expression $x_i < s$. This means that the number of genes analyzed is not exactly the same for all GRNs—we therefore normalized the distribution of perturbation effects by the number of genes that are included in the analysis (i.e., those not lowly-expressed; see S9 Fig).

Finally, we compared the distributions of incoming and outgoing perturbation effects using the Kolmogorov-Smirnov test as implemented in scipy (``scipy.stats.ks_2samp``) [42]. This is a nonparametric test for equality of distribution between two samples, which measures the maximum difference between cumulative distribution functions. To select the synthetic GRNs which are closest to the real data, we rank GRNs by largest KS p-values with

each distribution (incoming and outgoing), then either sum these ranks to get an overall ranking of networks (as in S11 Fig), or find the smallest rank $r$ such that $k$ GRNs are in the top $r$ of all GRNs compared to both distributions (as in Fig 5). These approaches generally agree on rankings, and nominate the same GRN as closest to the real data; we use this GRN for additional analysis. Its parameters are as follow: $k = 10, r = 4, \delta_{out} = 30, \delta_{in} = 3, w = 400$.

**4.3.3. Comparing with external data.** Inspired by the recent CausalBench inference challenge [11], we assessed the concordance of pairs of genes ordered by their perturbation effect $p$-values (from the K562 Perturb-seq dataset [9]) with pairwise links from three external data sources: the STRING database of physical and functional protein-protein interactions [43], the CORUM database of protein complexes [44], and a K562 ChIP-seq dataset curated by the CausalBench authors using data from ENCODE [45] and the Chip-Atlas [46]. As in the CausalBench truth set, we considered only pairs of genes with pairwise KO effects which were significant at a genome-wide FDR threshold of 0.05, to avoid false-positives in the truth set. Protein links were added as undirected edges (from gene "i" to gene "j" and vice versa) in the truth set, and ChIP-seq links were added as directed edges, considering only pairs of genes which were in our analysis subset of 5,247 genes, as defined above.

We ranked (directed) pairs of perturbed ($i$) and responding ($j$) genes in two ways, using the provided FDR-corrected Anderson-Darling p-values. One way is to simply use the raw p-value, $p_{ij}$, for each gene pair ($p_{KO}$ in S18 Fig); the other was to scale the p-value by the number, $n_i$, of genome-wide significant perturbation effects outgoing from gene $i$, i.e. $p'_{ij} = p_{ij}/n_i$ ($p'_{KO}$ in S18 Fig). The intuition for this latter scaling is that it biases the ranking to encourage edges to be drawn out from putative master regulators (with many outgoing perturbation effects).

## 4.4. Gene programs

**4.4.1. Truncated singular value decomposition.** We used truncated singular value decomposition (TSVD) to cluster genes into "programs" based on their expression profiles in cells from both perturbed and unperturbed settings, using the `TruncatedSVD` function from `scikit-learn` [47]. Briefly, TSVD is an algorithmic modification of singular value decomposition (SVD), which produces orthogonal singular vectors corresponding to the directions of maximum variance in the input data. In TSVD, only the top $k$ vectors are computed, which results in faster computational runtimes for our analysis.

We assembled separate input datasets consisting of perturbed and unperturbed cells for both synthetic data and using downsamples of the experimental Perturb-seq data. For the synthetic data, we simulated 75,328 single cells from baseline conditions by forward simulation from the expression fixed point of the GRN, sampling cells from every forward time step. We also simulated the expression of an identical number of cells under perturbed conditions, modeling the split of cells after the real Perturb-seq study: roughly 8.1% of the cells corresponded to baseline conditions, and the remainder were assigned uniformly at random to a knockout condition for each of the 2,000 genes in the GRN (this corresponds to 35 cells per KO on average). We do not filter out lowly expressed genes for this analysis.

For the real data, we used single-cell expression data of the 5,247 genes in our data subset from all 75,328 control cells as measurements of the GRN in unperturbed conditions. Then, to avoid effects from varying the size of the input cell population, we performed two independent (random) downsamples of the entire experiment to the same number of cells as measurements of the GRN in perturbed conditions.

With each of these input datasets $X$, we normalized each gene to have zero mean and unit variance, and then performed TSVD to compute the top $k = 200$ dimensions of expression variability. This resulted in singular matrices for cells ($U$) and genes ($V$), and a diagonal matrix of singular values, $S$. The product of these matrices approximates the input:

$$X \approx USV^{\top}$$

and we used the gene loadings (columns $v$ of the gene singular matrix $V$) to define gene programs. Each "program" corresponds one-to-one with one of the gene singular vectors $v$, and is the set of 200 genes with the largest squared entries of $v$.

**4.4.2. Similarity across datasets.** We assess the similarity of gene programs from two different experiments in two distinct ways: one using the set of genes which constitutes each program, and the other using the singular vector used to define it. When comparing programs $\{p_i\}$ from one (reference) experiment to programs $\{p'_j\}$ from another experiment, we report the maximum overlap between each program $p_i$ in the reference set to *any* program $p'_j$ in other set—that is,

$$\text{overlap}(p_i, \{p'_j\}) = \max_j |p_i \cup p'_j|$$

which quantifies the extent to which each program is reproduced by the other experiment. When comparing gene singular vectors $V = \{v_i\}$, $V' = \{v'_i\}$ from the two experiments, we make use of the fact that the SVD of their dot product is a well-characterized mathematical procedure called canonical correlation analysis (CCA) [48]. The top $k$ components of this decomposition are called canonical variables, and they each represent the axes of rotation which maximize correlation between variables in the input data. Here, we report the canonical correlation (singular values from the second SVD step) for the first 200 canonical variables, to quantify the extent to which the lower-dimensional representations of the inputs are consistent with one another.

## Supporting information

**S1 Fig: Relationship between $\delta$ and node degree in synthetic networks**. Example in- and out-degree distribution for four of the 1,920 GRNs simulated for the study. The networks have $n = 2,000$ genes, $r = 16$, $k = 1$, $w = 1$, and $\delta_{in}$ and $\delta_{out}$ either equal to 10 or as specified by the subpanel legend. The left columns show the relationship between $\delta_{in}$ and the distribution of incoming edges per gene, and the right columns show the relationship between $\delta_{out}$ and the distribution of outgoing edges per gene. The top and bottom rows display different views of the same data; the $x$-axis values in the top row are soft clipped at 100, and the $x$- and $y$-axes in the bottom row are log-scaled.
(TIFF)

**S2 Fig: Mediated effects outnumber direct effects at most magnitudes**. Same as Fig 3D, but with distances binned by whether pairs of genes are connected by an edge (distance 1, a "direct effect"), any path (distance greater than 1, a "mediated effect"), or no path at all ("null"). Note also that the $y$-axis is the count of gene pairs with a perturbation effect of at least the magnitude given on the $x$-axis—that is, the distribution shown is a non-normalized inverse CDF. Gene pairs are pooled from the 50 example GRNs in Fig 3.
(TIFF)

**S3 Fig: Modularity term differentiates within- and between-module effects**. Same as Fig 3E, with within-module perturbation effects in red and between-module perturbation effects in blue. Here, networks are chosen so as to highlight the effect of the modularity term $w$. Each pair of blue and red tracelines is distribution of the within- (red) or across-module perturbation effects a single GRN. The generating parameters for these GRNs vary $w$ (see legend) but hold other parameters constant, as follow: $p = 1/4$, $k = 50$, $\delta_{in} = 10$, $\delta_{out} = 10$.
(TIFF)

**S4 Fig: Network generating parameters affect the number of KO effects**. Counts of the number of perturbation effects per gene in the GRN with $|\log_2 FC| \geq 0.1$ in each synthetic GRN, as a function of network generating parameters. Each panel shows all 1,920 GRNs as individual points, stratified by parameter values. Each distribution is annotated with its mean over GRNs (diamond points). We observe a similar direction of effect for each parameter as with the statistics presented in Fig 4.
(TIFF)

**S5 Fig: Network generating parameters affect the stability of the fixed point**. Counts of the number of genes which are hub targets in each synthetic GRN, as a function of network generating parameters, and stratifying by whether the expression equilibrium point of the synthetic GRN is stable (**Methods**). In all, 1,693 of the 1,920 GRNs (88.2%) reach an expression equilibrium through forward simulation of the SDE which is a stable fixed point of the corresponding ODE. These GRNs tend to be sparse (lower $1/p$), modular (higher $w$), and have more hub regulators (lower $\delta_{out}$), consistent with the direction of effect on the number of strong KOs and key target genes (as in Fig 4).
(TIFF)

**S6 Fig: No interaction between sparsity term and other network generating parameters**. Counts of the number of genes which are strong knockouts (KOs, top row) and key target genes (bottom row) in each synthetic GRN. Each panel shows all 1,920 GRNs as individual points, across values of network generating parameters ($x$-axes), with additional stratification by the sparsity term $1/p$. Each distribution is annotated with its mean over GRNs in each bin (diamond points). There is no obvious visual evidence for interactions between the parameters.
(TIFF)

**S1 Data: K-S statistics for synthetic GRNs**. Network generating parameters for all GRNs in the study: number of genes $n$, sparsity $r$, groups $k$, modularity $w$, degree dispersion $\delta_{in}$ and $\delta_{out}$. K-S test statistic for perturbation effects from each network, when compared with the distribution of incoming ("tg_ks0") or outgoing ("ko_ks0") perturbation effects observed in real data (as in Fig 5).
(CSV)

**S7 Fig: No interaction between out-degree term and other network generating parameters**. Counts of the number of genes which are strong knockouts (KOs, top row) and key target genes (bottom row) in each synthetic GRN. Each panel shows all 1,920 GRNs as individual points, across values of network generating parameters ($x$-axes), with additional stratification by the out-degree term $\delta_{out}$. Each distribution is annotated with its mean over GRNs in each bin (diamond points). There is no obvious visual evidence for interactions between the parameters.
(TIFF)

**S8 Fig: Summary of regression models for effects of network parameters on perturbations**.
Coefficients from regressing the logit-transformed fraction of genes which are hub knockouts
(top) or target genes (bottom) on network generating parameters. Error bars denote 95% confidence intervals for the regression coefficients. Model summaries can be found in S1 Table
(hub knockouts) and in S2 Table (target genes).
(TIFF)

**S1 Table: Summary of regression results (fraction of genes which are hub knockouts).**
(CSV)

**S2 Table: Summary of regression results (fraction of genes which are hub target genes).**
(CSV)

**S9 Fig: Number of genes included in analysis of each simulated GRN**. Distribution of the
number of non-lowly expressed genes included in the matching analysis in Fig 5 (see **Methods**). The top left panel shows the distribution over all 1,920 GRNs; each other subpanel
shows the conditional distributions for each generating parameter. We note an interaction
between the number of regulators per gene and the number of genes included in downstream
analysis.
(TIFF)

**S10 Fig: No interaction between network generating parameters and fit to experimental
data**. As in Fig 5C–E, we show the relationship between pairs of network generating parameters and goodness of fit to the cumulative distribution of perturbation effects from experimental Perturb-seq data. Each GRN (one point in every subpanel) is colored by its ranked fit
to data: the synthetic GRNs are ranked separately by Kolmogorov-Smirnov $p$-value for incoming and outgoing perturbation effects, then the sum of these two ranks is used to produce an
overall ranking. Intense red color indicates better ranked fit to data, and intense blue color
indicates a worse ranking.
(TIFF)

**S11 Fig: Closest $k$ simulated GRNs to real data**. Replication of Fig 5C–G, changing the number of plotted GRNs. Distribution of network generating parameters for the $k$ GRNs that
are best matched to Perturb-seq data (by K-S $p$-value rank for the number of incoming and
outgoing effects; see **Methods**).
(TIFF)

**S12 Fig: Resampled expression models for the four closest GRNs to real data**. Replication
of Fig 5A and 5B for the four GRNs closest to real Perturb-seq data. For each GRN, we resample the parameters of the expression model ($\alpha, \ell, \beta$; see **Methods**) 20 times and recompute
the distribution of KO effects (outgoing, left panel; incoming, right column). The GRNs numbered 1, 2, 3, and 4 in Fig 5 are respectively numbered 647, 1014, 678, and 677 here. GRN
number 647 (1) is the GRN used in Figs 6 and 7.
(TIFF)

**S13 Fig: Baseline expression influences the number of outgoing and incoming perturbation effects**. In the subsetted data from Replogle et al. [9], and in the focal GRN from Fig 6,
we show the relationship between mean expression (in control cells in the experimental data;
top panels) or steady-state expression (in the synthetic GRN; bottom panels) and the number
of incoming (left) or outgoing (right) perturbation effects. Baseline expression relates to both

of these quantities in both data sets: the relationships are stronger in the experimental data in part due to limits on detection power (especially important for incoming effects).
(PNG)

**S14 Fig: Distribution of threshold log fold-change in synthetic GRNs**. Critical values of $|\log_2 \text{FC}|$ used to match the discovery rate of real Perturb-seq data in Fig 5. The top left panel shows the distribution over all 1,920 GRNs; each other subpanel shows the conditional distributions for each generating parameter. We note a relationship between the number of regulators per gene ($r$), hub regulatory architecture ($\delta_{out}$), and this critical value.
(TIFF)

**S15 Fig: Coexpression is more often due to coregulation than edges**. In the focal GRN from Fig 6, we show a histogram of coexpression values split by whether pairs of genes share an edge ("A regulates B, or B regulates A"), share a regulator ("A and B are coregulated"), or have another relationship (left panel). Similarly, for perturbation effects, we show the distribution split by whether pairs of genes share an edge ("A regulates B"), a path of distance 2 ("A indirectly regulates B"), or another relationship (right panel). At nearly all levels of coexpression, coregulation is more common than direct regulation. Meanwhile, direct regulation is more common than indirect regulation for the largest perturbation effects—note that the range of KO effects is clipped as in Fig 6.
(TIFF)

**S16 Fig: Baseline coexpression and perturbation effects are uncorrelated in Perturb-seq data**. Same as Fig 6E, using data from our analysis subset of Replogle et al. 2022 [9]. Gene coexpression (x-axis) is the unsigned Pearson correlation between normalized single-cell gene expression data from unperturbed cells (clipped at $|r| = 0.1$). Perturbation effects (y-axis) are pairwise log-transformed Anderson-Darling p-values for differences in gene expression distribution between perturbed and unperturbed states (clipped at $-\log_{10}(p) = 10$). Rank correlation (Spearman's $\rho$) is computed on the transformed but not clipped values of these two statistics.
(TIFF)

**S17 Fig: Graph properties inform the extent to which perturbation effects more strongly enrich for edges than co-expression**. Performance of perturbation effects ($x$-axis, $|\log_2 \text{FC}|$) and co-expression ($y$-axis, $r^2$ between genes) in identifying edges. As a summary performance measure, we compute the average precision (AP) score during binary classification of pairs of genes as being connected an edge (ignoring direction). All 1,920 networks in the study are shown in both panels, and colored by parameters of interest: the sparsity parameter ($r$, left), the group affinity parameter ($w$, middle), and the out-degree parameter ($\delta_{out}$, right). Across networks, sparsity, modularity, and degree uniformity degrade the performance of coexpression values, but enhance the performance of perturbation effects; but in every network, perturbation effects outperform coexpression (all points are below the dashed grey line, $y = x$).
(TIFF)

**S18 Fig: Graph properties make a useful prior for edge prediction**. Performance of perturbation effect $p$-values (blue line, $p_{KO}$) and re-weighted perturbation effect $p$-values (orange line, $p'_{KO} = p_{ij}/n_i$) in identifying pairs of genes with established protein-protein interactions, shared protein complex membership, or ChIP-seq links (**Methods**). In the prior expression, $p_{ij}$ is the $p$-value for the response of gene $j$ to a perturbation of gene $i$, and $n_i$ is the number

of genome-wide significant effects (at FDR 0.05) that perturbation of gene $i$ has on all other genes in the dataset.
(TIFF)

**S19 Fig: True groups in the synthetic GRN are represented among gene programs**. In the focal GRN from Fig 7, we show the overlap between each of the true groups ($k$ = 10, shown as points in each of the bins on the $y$-axis) and its closest matching program (maximum overlap across all 50 gene sets, values shown on the $x$-axis). Points corresponding to the same true group are connected with a line spanning across $y$-axis bins. There is similar representation of all of the groups among the learned gene programs, regardless of input data type.
(TIFF)

**S20 Fig: Program replication depends on the number of cells**. Same as Fig 7C and D—instead of taking downsamples of unperturbed cells from Replogle et al., 2022, we downsample the entire experiment to various study sizes. Here, the "entire experiment" is the normalized expression measurements of 5,247 genes in 932,593 control and intervened-upon cells which received one of the 5,247 perturbations in our analysis subset (**Methods**). We compare singular vectors (left) and programs (right) from the resulting downsamples of the entire experiment, as well as the subsets from Fig 7C and D. We note that the 75,328 control cells replicate the programs from the entire dataset comparably to 30,000 cells from the entire experiment.
(TIFF)

**S21 Fig: Program replication depends on the magnitude of intrinsic noise**. Same as Fig 7A and B for different levels of noise. We repeat CCA and analysis of gene programs as in Fig 7 (see **Methods**), varying the level of intrinsic noise ($s$). At low levels of noise (small $s$), replicates from perturbed conditions are much more similar to one another than to the unperturbed data. At high levels of noise (large $s$), the perturbed data are more similar by canonical correlation to the unperturbed data than to the replicate perturbed data; but programs derived from each of the singular vectors are equivalently reproducible across conditions.
(TIFF)

## Acknowledgments

We would like to thank Mineto Ota, Emma Dann, Courtney Smith, Garyk Brixi, Josh Weinstock, and members of the Pritchard Lab at Stanford University for helpful comments and discussion related to this work.

## Author contributions

**Conceptualization:** Matthew Aguirre, Jeffrey P. Spence, Guy Sella, Jonathan K. Pritchard.

**Data curation:** Matthew Aguirre.

**Formal analysis:** Matthew Aguirre.

**Funding acquisition:** Matthew Aguirre, Guy Sella, Jonathan K. Pritchard.

**Investigation:** Matthew Aguirre.

**Methodology:** Matthew Aguirre, Jeffrey P. Spence, Guy Sella, Jonathan K. Pritchard.

**Project administration:** Matthew Aguirre, Jeffrey P. Spence, Guy Sella, Jonathan K. Pritchard.

**Resources:** Jonathan K. Pritchard.

**Software:** Matthew Aguirre.

**Supervision:** Jeffrey P. Spence, Guy Sella, Jonathan K. Pritchard.

**Validation:** Matthew Aguirre.

**Visualization:** Matthew Aguirre.

**Writing – original draft:** Matthew Aguirre, Jeffrey P. Spence, Guy Sella, Jonathan K. Pritchard.

**Writing – review & editing:** Matthew Aguirre, Jeffrey P. Spence, Guy Sella, Jonathan K. Pritchard.

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
