## [Decision Letter · Decision Letter 0]

23 Dec 2024

PCOMPBIOL-D-24-01817

Gene regulatory network structure informs the distribution of perturbation effects

PLOS Computational Biology

Dear Dr. Aguirre,

Thank you for submitting your manuscript to PLOS Computational Biology. After careful consideration, we feel that it has merit but does not fully meet PLOS Computational Biology's publication criteria as it currently stands. Therefore, we invite you to submit a revised version of the manuscript that addresses the points raised during the review process.

Please submit your revised manuscript within 60 days Feb 22 2025 11:59PM. If you will need more time than this to complete your revisions, please reply to this message or contact the journal office at ploscompbiol@plos.org. Please include the following items when submitting your revised manuscript:

We look forward to receiving your revised manuscript.

Kind regards,

Emily Miraldi, Ph.D.

Guest Editor

PLOS Computational Biology

Sushmita Roy

Section Editor

PLOS Computational Biology

**Journal Requirements:**

At this stage, the following Authors/Authors require contributions: Matthew Aguirre, Jeffrey P. Spence, Guy Sella, and Jonathan K. Pritchard. Please ensure that the full contributions of each author are acknowledged in the "Add/Edit/Remove Authors" section of our submission form.

4) Your manuscript is missing the following section: Results.  Please ensure all required sections are present and in the correct order. Make sure section heading levels are clearly indicated in the manuscript text, and limit sub-sections to 3 heading levels. An outline of the required sections can be consulted in our submission guidelines here:

5) Please upload all main figures as separate Figure files in .tif or .eps format. For more information about how to convert and format your figure files please see our guidelines: 

6) We have noticed that there is a reference to "Supplementary Data 1"  in your manuscript. However, there is no corresponding file uploaded to the submission. Please upload it as a separate file with the item type 'Supporting Information'.

7) We notice that your supplementary Figures, and Tables are included in the manuscript file. Please remove them and upload them with the file type 'Supporting Information'. Please ensure that each Supporting Information file has a legend listed in the manuscript after the references list.

8) We note that your Data Availability Statement is currently as follows: "All relevant data are within the manuscript and its Supporting Information files." Please confirm at this time whether or not your submission contains all raw data required to replicate the results of your study. Authors must share the “minimal data set” for their submission. PLOS defines the minimal data set to consist of the data required to replicate all study findings reported in the article, as well as related metadata and methods (https://journals.plos.org/plosone/s/data-availability#loc-minimal-data-set-definition).

9) Please amend your detailed Financial Disclosure statement. This is published with the article. It must therefore be completed in full sentences and contain the exact wording you wish to be published.

1) State what role the funders took in the study. If the funders had no role in your study, please state: "The funders had no role in study design, data collection and analysis, decision to publish, or preparation of the manuscript.".

10) Please ensure that the funders and grant numbers match between the Financial Disclosure field and the Funding Information tab in your submission form. Note that the funders must be provided in the same order in both places as well. Currently, the order of the grants is different in both places.

Please indicate by return email the full and correct funding information for your study and confirm the order in which funding contributions should appear. Please be sure to indicate whether the funders played any role in the study design, data collection and analysis, decision to publish, or preparation of the manuscript.

**Reviewers' comments:**

Reviewer's Responses to Questions

Reviewer #1: The paper proposed a method to simulate perturbation on GRNs. My major concern:

1. There is a big gap between simulation and real-world application. Many details on the application on [9] is missing. It is hard to evaluation the method's performance in real-world applicaiton.

2. The method sample many random GRNs, and simulated them and see their behavior. However, the results depend on how good the sampling is. But it is hard to know how many samples we need to have meaningful results. What if all GRN samples are not correct, then all the observations would be wrong.

3. How model parameters betas are determined. Are they also random? If so, then there are to many random effects in the model. All the conclusion could be different if we run the method again.

Reviewer #2: The manuscript by Aguirre et al., “Gene regulatory network structure informs the distribution of perturbation effects,” introduces a method for generating gene regulatory networks (GRNs) that allows for the control of network properties through generation parameters. This is followed by a stochastic differential equations model to simulate perturbation effects on gene expression. The authors provide a thorough analysis of how network properties influence gene perturbation effects, demonstrating that synthetic networks generated by their method can recapitulate certain characteristics observed in experimental Perturb-seq data. Notably, the authors compare the utility of co-expression and perturbation data for GRN inference tasks, concluding that perturbation data is more effective for inferring regulatory edges, while both perturbed and unperturbed data identify similar gene programs.

The manuscript is well-written, easy to follow, and the dataset and code have the potential to be highly valuable for advancing the understanding and modeling of GRNs.

I have three comments that I believe could help clarify key points and enhance the study's impact.

1. Synergistic Regulation in GRNs

Synergistic regulation is a critical feature of biological GRNs. In the manuscript, the authors model the transcript synthesis rate as a sigmoid function of the combined linear effect of regulator expression. Could the authors clarify whether their model accounts for synergistic effects among regulators? If this aspect is not included in the current model, it would be important to address this limitation in the Discussion section.

2. Bimodal Distribution of Hub Target Genes (Figure 4)

In the text for Figure 4A (Lines 220-221), the authors note, “However, in a subset of dense networks, most genes in the network are identified as hub target genes.” In fact, in other panels of Figure 4, the number of hub target genes often displays a similar bimodal distribution, independent of the network properties. This is not observed for hub KO genes. Could the authors provide an explanation for this pattern? Are there unique characteristics in networks where nearly all genes are identified as hub target genes?

3. Data and Code Availability

I strongly recommend that the authors make their code and synthetic network data publicly available, including proper references to these resources in the manuscript. While the authors state in Lines 67-68 that their tools are available on GitHub, no URL is provided. Additionally, the “Data and Code Availability” section indicates, “All relevant data are within the manuscript and its Supporting Information files.” However, I was unable to locate the synthetic network data (network structure and simulated expression) in Supporting information files. As the authors noted in the Discussion, their technique could significantly benefit model development and benchmarking. Making these datasets publicly available would enable further studies of GRN properties and inference, greatly enhancing the utility and impact of this work.

Reviewer #3: In this manuscript, the authors present a novel analysis of perturb-seq data through the lens of gene regulatory networks. By comparing real-world perturb-seq data to synthetic perturbation data, they offer a fresh perspective on interpreting noisy single-cell data. The manuscript is well-structured and clearly written. I recommend to accept this manuscript for publication.

**Have the authors made all data and (if applicable) computational code underlying the findings in their manuscript fully available?**

Reviewer #1: **No: **

Reviewer #2: **No: **GitHub URL for the tool is not included in the manuscript.

Reviewer #3: Yes

PLOS authors have the option to publish the peer review history of their article (what does this mean?). If published, this will include your full peer review and any attached files.

Reviewer #1: No

Reviewer #2: No

Reviewer #3: **Yes: **Zhana Duren

**Figure resubmission:**
---

## [Decision Letter · Decision Letter 1]

18 Apr 2025

PCOMPBIOL-D-24-01817R1

Gene regulatory network structure informs the distribution of perturbation effects

PLOS Computational Biology

Dear Dr. Aguirre,

Thank you for submitting your manuscript to PLOS Computational Biology. After careful consideration and in-depth discussion among the editors, we feel that while it has merit it does not fully meet PLOS Computational Biology's publication criteria as it currently stands. In addition to unsatisfactory response to reviewer 1, the editors have additional concerns. We invite you to submit a revised version of the manuscript that addresses the points raised during the review process.

However, we understand that the paper has been under review for a long period of time. If the authors choose to not address these concerns, the paper will be rejected with the option of transfer to PLOS one. We note that final acceptance is still subject to authors fully addressing the reviewer and editor comments. We understand this is a non-trivial ask and therefore we offer the transfer option. The editors are below.

Please submit your revised manuscript within 60 days Jun 18 2025 11:59PM. If you will need more time than this to complete your revisions, please reply to this message or contact the journal office at ploscompbiol@plos.org. Please include the following items when submitting your revised manuscript:

We look forward to receiving your revised manuscript.

Kind regards,

Emily Miraldi, Ph.D.

Guest Editor

PLOS Computational Biology

Sushmita Roy, Ph.D.

Section Editor

PLOS Computational Biology

Shihua Zhang

Section Editor

PLOS Computational Biology

**Additional Editor Comments:**

In mammalian systems, ground truth knowledge of GRNs is lacking, and generating synthetic networks to explore the relationship between network parameters and experimentally measurable GRN outputs (e.g., gene expression responses to systematic gene KO) is a valuable contribution and will be of interest to the computational biology community. However, there are opportunities to improve the rigor and communication of the work. Please see below.

Major Concerns:

The DREAM literature showed that GRNs inference methods performing well on synthetic GRNs have failed to translate that performance to real datasetsTo demonstrate impact, authors would need to show how leveraging insights from their synthetic GRN analysis led to improvements to a GRN inference method that could then be evaluated using standard performance criteria (precision-recall of a gold standard network in K562, e.g., leveraging leave-out form the CRISPR screen, rich TF ChIP-seq from ENCODE...)? However, this is a nontrivial ask and beyond the scope of a revision (or the goals of the study as argued in the rebuttal).There are a variety of GRN definitions in the literature. Please clearly define that your GRN allows regulatory interactions between any genes in the network. In contrast, another GRN definition limit to regulatory interactions from protein TFs to genes. You are clearly not using this TF-centric GRN definition, as your network generating algorithm allows for regulatory interactions between all genes (direct or indirect) and this makes sense given eventual goal to compare to the K562 screen. However, there are a couple of places in the text where you muddy the water by invoking “TFs”. Please replace “TFs” with “regulators” in Fig. 2B, for example. More broadly, it will be valuable to discuss limitations of your GRN definition in Discussion, as your ODE model assumes that the transcript of any gene can regulate transcription of any other gene, when biologically a gene regulatory network involves: mRNA � protein � protein-protein interactions � TFs � gene expression. This is a limitation of your GRN network model and it would be of value to the readership to discuss. (Relatedly, how might this more complex biological process be embedded into your beta coefficients or addressed experimentally and/or computationally?)Fig. 5C shows properties of the 100 synthetic GRNs matching the KO perturbation distributions in the K562 screen. This is a key result. How were K-S statistics on 5A and 5B distributions combined (average rank)? Was there a natural break point in the combined similarity scores that would be more appropriate than 100? How robust are the results to the top 100 cutoff? What if you look at the top 5-10 most similar synthetic GRNs, do conclusions change? One could easily add data points of different colors corresponding to top 1, top 5, top 10, top 50 on 5C… Could be helpful to model similarity of synthetic networks’ to K562 distributions as a function of synthetic network parameters, to further pinpoint those parameters most impacting similarity – but likely beyond the scope of this manuscript.The parameterization of the stochastic ODEs is random. A major goal of the study is to connect the topology-driving parameters used in network topology generation to simulations from the stochastic ODEs. Across random stochastic ODE parameterizations for the same network topology, how reproducible are the KO perturbations and resulting CDFs of % genes perturbed per KO and % KOs effecting genes (e.g., Fig. 5A, B) across random stochastic ODE parameterizations? To relate network topology parameters to KO simulations, it is important to establish that the CDFs are robust to random parameter sets. (Or to average across random parameterizations, if the KO simulations are sensitive to the random parameterization.) This is something that could be examined, say, with 5 or so random parameterizations of the 4 synthetic networks whose KO distributions were most similar to the K562 screen results. 

Minor concerns:

In section 2.5, paragraph 1, genes whose equilibrium expression was below 10^-4 were removed from GRNs, but later (Fig. 5) the major network stats reported are based on the fraction of genes impacted by KO. How did that effect the denominator in Fig. 5A, B fractions? E.g., are you looking at only 600 genes or closer to 2000? Would be helpful to show a distribution of % genes above noise if they were a significant fraction of the 2000 total.In section 2.6 / Fig. 5, to match perturb-seq data, genes were thresholded at the top 3% of absolute log2-FCs. On average, what log2-FC value did this correspond to? (10% (similar to cutoff used in earlier figures), something much smaller 10^-3?) Would be helpful to show the distribution FC cutoffs corresponding to 3%, to give readers a sense of effect size in model.Fig. 1 – add y axis quantification to 1CFig. 2B – It’s hard to get a visceral sense for the out-degree CV, would be beneficial to show some out-degree distributions for GRNs at particular delta_out values. For example, for out-degree CV = 5. What max out-degree does that correspond to? (200 or 2000 targets per TF?) Or better yet, would be helpful to show distributions of the max and mode out or in degrees for delta_out and delta_in parameters. Also, please improve the figure legend so that abbreviations (e.g., CV) are defined in legend, in addition to text.Fig. 3 analysis shows results for 50 GRNs. How were these 50 GRNs selected? Why not use all ~2000 or so?Fig. 6A/B violin plot blue is hard to distinguish from gray boxplot interquartile ranges. Fig. 7, add to legend how many genes per program are shown. In Methods, the K562 screen 100 genes per program are used, but Gene count is 200 in 7B, so not sure what number was used. Fig. S1 would benefit from grid lines and a mark at log2(FC) = .1, the cutoff used to define hub KOs and their targets. How were 50 GRNs selected for inclusion in the figure? (Why not show all?)Fig. S3 and S4, please include full descriptions of the results shown rather than having reader refer back to Fig. 4 legend.

**Reviewers' comments:**

Reviewer's Responses to Questions

**Comments to the Authors:**

Reviewer #1: I want to thank the authors for the detailed explanation for my concerns.

I still have some questions about how to connect synthetic network to real world data.

1. how you identify a synthetic GRN that is similiar to the pertub-seq data. In which criteria?

2. How you pick the subset of the perturbations? What are those genes?

3. Which kind of inference the method proposed can do? Can they infer TF-gene regulation?

Reviewer #2: The authors have adequately addressed my comments.

Reviewer #3: recommend to accept for publication.

**Have the authors made all data and (if applicable) computational code underlying the findings in their manuscript fully available?**

Reviewer #1: None

Reviewer #2: Yes

Reviewer #3: Yes

PLOS authors have the option to publish the peer review history of their article (what does this mean?). If published, this will include your full peer review and any attached files.

Reviewer #1: No

Reviewer #2: No

Reviewer #3: **Yes: **Zhana Duren

**Figure resubmission:**
---

## [Decision Letter · Decision Letter 2]

3 Aug 2025

Dear Mr. Aguirre,

We are pleased to inform you that your manuscript 'Gene regulatory network structure informs the distribution of perturbation effects' has been provisionally accepted for publication in PLOS Computational Biology.

Best regards,

Sushmita Roy, Ph.D.

Section Editor

PLOS Computational Biology

Sushmita Roy

Section Editor

PLOS Computational Biology

Reviewer's Responses to Questions

**Comments to the Authors:**

Reviewer #1: I have no futher questions.

**Have the authors made all data and (if applicable) computational code underlying the findings in their manuscript fully available?**

Reviewer #1: Yes

PLOS authors have the option to publish the peer review history of their article (what does this mean?). If published, this will include your full peer review and any attached files.

Reviewer #1: No

---

## [Editor Report · Acceptance letter]

PCOMPBIOL-D-24-01817R2

Gene regulatory network structure informs the distribution of perturbation effects

Dear Dr Aguirre,

I am pleased to inform you that your manuscript has been formally accepted for publication in PLOS Computational Biology. Your manuscript is now with our production department and you will be notified of the publication date in due course.

With kind regards,

Aiswarya Satheesan
